# Impact of temperature on the role of Criegee intermediates and peroxy radicals in dimers formation from β-pinene ozonolysis

Yiwei Gong[1,2], Feng Jiang[2], Yanxia Li,[2] Thomas Leisner[2,3], and Harald Saathoff[2]

[1]Department of Atmospheric and Oceanic Sciences, School of Physics, Peking University, Beijing, China
[2]Institute of Meteorology and Climate Research, Karlsruhe Institute of Technology, Karlsruhe, Germany
[3]Institute of Environmental Physics, Heidelberg University, Heidelberg, Germany

*Correspondence to*: Yiwei Gong (yiwei.gong@kit.edu) and Harald Saathoff (harald.saathoff@kit.edu)

**Abstract.** Stabilized Criegee intermediates (SCIs) and organic peroxy radicals ($RO_2$) are critical in atmospheric oxidation processes and secondary organic aerosol (SOA) formation. However, the influence of temperature on their corresponding reaction mechanisms in SOA formation is unclear. Through utilizing formic acid as SCIs scavenger and regulating the ratio of hydroperoxyl radials ($HO_2$) to $RO_2$ ($[HO_2]/[RO_2]$) from ~0.3 to ~1.9 using different concentrations of CO, the roles of $RO_2$ and SCIs in SOA formation were investigated from 248 K to 298 K, particularly for dimers formation in β-pinene ozonolysis. The SOA yield increased by 21% from 298 K to 273 K, while decreased by 40% from 273 K to 248 K. Both changing $[HO_2]/[RO_2]$ and scavenging SCIs significantly affect SOA yield and composition. SCIs reactions accounted for more than 40% of dimers and SOA mass formation for all temperatures. Increasing $[HO_2]/[RO_2]$ inhibited dimers and SOA formation, and this inhibition became larger with decreasing temperature. Compared to low $[HO_2]/[RO_2]$ (0.30–0.34), the dimers abundance at high $[HO_2]/[RO_2]$ (1.53–1.88) decreased by about 31% at 298 K and 70% at 248 K. $[HO_2]/[RO_2]$ has a specific impact on SCIs-controlled dimers at lower temperatures by influencing especially the $C_9$-SCIs reactions with $RO_2$. The dimers formed from $C_9$-SCIs reactions with $RO_2$ were estimated to decrease by 61% at high $[HO_2]/[RO_2]$ compared to low $[HO_2]/[RO_2]$ at 248 K. The high reactivity and substantial contribution to SOA of β-pinene-derived SCIs at lower temperatures observed in this study suggest that monoterpene-derived SCIs reactions should be accounted for in describing colder regions of the atmosphere.

## 1 Introduction

Secondary organic aerosols (SOA) have received considerable attention over the past decades due to their critical role in air quality and climate change (Hallquist et al., 2009; Kanakidou et al., 2005). Although significant progress has been made in understanding and modeling SOA formation and composition, the impact of temperature on the mechanism of SOA formation is still not well understood, especially for colder conditions (≤ 0 ℃) (Porter et al., 2021). Several studies reported higher SOA yields at lower temperatures in α-pinene oxidation (Jonsson et al., 2008; Pathak et al., 2007, 2008; Saathoff et al., 2009). In recent years, more attention has been paid to the temperature impact on aerosol constituents and physicochemical properties. Ye et al. (2019) and Simon et al. (2020) reported highly oxygenated molecules (HOMs) were less abundant at lower temperatures in α-pinene oxidation due to the positive temperature dependence of autoxidation reaction (Praske et al., 2017), however, the reduction of the saturation vapor pressure at lower temperatures counteracted the chemical effect on new particle formation. Kristensen et al. (2017) reported suppressed formation of dimers at subzero temperatures. Huang et al. (2018) studied the interactions between particle composition and viscosity at 223 K. Gao et al. (2021, 2023) investigated the temperature effect on the composition and volatility of aerosols from β-caryophyllene oxidation. These studies indicate that besides the impact on volatilities and partitioning, temperature impacts the chemical reaction mechanism and product formation. Although the temperature impact on HOMs formation was studied, the understanding of the temperature impact on the formation pathways of other important SOA constituents such as dimers is limited. This study tries to bring some new

insights into the reaction mechanisms of two kinds of reactive intermediates in the atmosphere: stabilized Criegee intermediates (SCIs) and organic peroxy radicals ($RO_2$) at lower temperatures, and their further impacts on dimers and SOA formation.

It has been proven that SCIs and $RO_2$ can react with other trace gas species and generate semi-volatile organic compounds (SVOCs) and low-volatile organic compounds (LVOCs) (Chhantyal-Pun et al., 2020a; Orlando and Tyndall, 2012). The structural diversity, short lifetime, and low concentration of SCIs and $RO_2$ make it challenging to study their fates in the

atmosphere. SCIs, formed from alkene ozonolysis, perform as an efficient oxidant for several trace species, e.g., $SO_2$, $NO_x$, carboxylic acids, carbonyl compounds, etc., contributing to the formation of inorganic and organic aerosol components (Cox et al., 2020; Percival et al., 2013). SCIs' reaction properties are structure-dependent, and although many researches have synthesized and studied simple SCIs containing $\leq 3$ carbon atoms, the reactivities of larger SCIs, such as monoterpene-derived and sesquiterpene-derived SCIs, are still vague (Lin and Chao, 2017). The bimolecular reactions of simple SCIs with $SO_2$,

carbonyl compounds, and water dimers have negative temperature dependences (Chhantyal-Pun et al., 2017; Onel et al., 2021; Smith et al., 2015; Wang et al., 2022). Lower temperatures could significantly promote the stabilization of SCIs and reduce the unimolecular decay rate of SCIs (Peltola et al., 2020; Robinson et al., 2022; Smith et al., 2016). This study will provide insight into whether this can lead to a different role of SCIs in SOA formation in winter and colder regions of the atmosphere. $RO_2$ radicals are vital in the atmospheric radical cycle, and reactions with hydroperoxyl radials ($HO_2$), $RO_2$, and NO are the

main reaction pathways of $RO_2$ in the atmosphere. The reactions with $HO_2$ and $RO_2$ are important for determining the fate of $RO_2$ in clean areas and urban areas with NO reduction. In recent years, autoxidation has been claimed to be a competitive reaction pathway for $RO_2$ radicals with a positive temperature dependence (Praske et al., 2017). The rate coefficient of $RO_2+HO_2$ is typically on the order of $10^{-11}$ $cm^3$ $molecule^{-1}$ $s^{-1}$ with a negative temperature dependence (Atkinson et al., 2006). As for $RO_2+RO_2$ reactions, which include the self- and cross-reactions of $RO_2$ radicals, the rate coefficients vary over a wide

range from $10^{-17}$ to $10^{-10}$ $cm^3$ $molecule^{-1}$ $s^{-1}$ (Berndt et al., 2018a; Tomaz et al., 2021). The impact of temperature on $RO_2+RO_2$ reactions was reported to be dependent on $RO_2$ structures, and there is no clear conclusion on the temperature influence on the rate coefficients and the product branching ratios of monoterpene-derived and sesquiterpene-derived $RO_2+RO_2$ reactions as for now (Atkinson et al., 2006). $[HO_2]/[RO_2]$ is not only critical in determining the fate of $RO_2$, but also important for evaluating whether the laboratory results can be compared with realistic situations. Atmospheric $[HO_2]/[RO_2]$ is usually larger

than 1, and the modeled global surface $[HO_2]/[RO_2]$ was reported as 2−9 in January and 0.75−2 in July (Peng et al., 2022). However, in simulation chamber or flow tube studies, without $HO_2$ sources, the $[HO_2]/[RO_2]$ could be significantly lower than 1, leading to $RO_2$ radicals primarily undergoing self- or cross-reactions.

In the past few years, the generation of dimers has attracted increasing attention due to the low volatilities of dimers, and is recognized as an important process in particle nucleation and growth (Donahue et al., 2012; Kristensen et al., 2013; Müller et

al., 2008). Some particle-phase reactions, including hemiacetal reactions of peroxides and carbonyls, noncovalent clustering of carboxylic acids, and aldol condensation reactions, could contribute to dimers formation (Kenseth et al., 2018; Yasmeen et al., 2010). However, these pathways were not able to adequately explain the dimers observed, and gas-phase reaction pathways were proposed to be important (DeVault and Ziemann, 2021; Hasan et al., 2021). The gas-phase formation mechanisms of dimers include reactions involving SCIs and $RO_2$ radicals, and clustering of carboxylic acids (Berndt et al., 2018a; Chen et

al.,2019; Kristensen et al., 2014; Valiev et al., 2019). Some field studies investigated the formation of dimers and showed potential temperature effects (Claudia et al., 2017; Yasmeen et al., 2010). The temperature dependence reported for dimers formation was contradictory due to the uncertainty of temperature impact on different formation pathways (Kristensen et al., 2020; Zhang et al., 2015). In this study, the performances of SCIs and $RO_2$ radicals in the generation of dimers and SOA were elaborately studied from 248 K to 298 K in β-pinene ozonolysis. Dimers' formation was particularly focused on, because of

their importance in particle generation and growth, as well as their role of an essential indicator for $RO_2$ and SCIs reactions. Monoterpenes are critical precursors for the generation of reactive intermediates and aerosols in the atmosphere. The oxidation of α-pinene has been broadly investigated, however, different isomers of monoterpenes have different reaction mechanisms

due to their molecule structures (Jenkin, 2004; Lee et al., 2006). As the second most abundant monoterpene, β-pinene with a global annual emission rate of 10−50 TgC contributes to about 20% of monoterpenes, and is regarded as a representative exocyclic monoterpene (Guenther et al, 2012; Sindelarova et al., 2014; Wiedinmyer et al, 2004). Docherty and Ziemann (2003) revealed the significant influence of $RO_2$ and SCIs reactions on aerosol formation from β-pinene ozonolysis. With a considerable SCIs yield of more than 0.4 (Nguyen et al., 2009), and slower $RO_2$ autoxidation (Ehn et al., 2014), β-pinene shows different oxidation mechanisms and products formation compared to α-pinene. This study investigates the role of SCIs and $RO_2$ in dimers and SOA formation from 248 K to 298 K in β-pinene ozonolysis, aiming to provide a more comprehensive understanding on the monoterpene oxidation in different regions of the atmosphere.

## 2 Experimental

### 2.1 Experiments

The experiments were conducted in the AIDA (Aerosol Interaction and Dynamics in the Atmosphere) simulation chamber at the Karlsruhe Institute of Technology (KIT). The AIDA chamber is a cylindrical aluminum vessel of 84.5 $m^3$ in volume. It was operated as a continuously stirred reactor with a mixing time of 1–2 min achieved by a fan located 1 m above the bottom of the chamber. The temperature inside the chamber was controlled at 298±0.3 K, 273±0.3 K, and 248±0.3 K during this work. β-pinene (99%, Alfa Aesar) was evaporated and added to the chamber with a flow of synthetic air. The initial mixing ratio of β-pinene at 298 K was 19.3±1.2 ppb, and the initial molecule concentration of β-pinene in each experiment was controlled to be similar, resulting in correspondingly lower mixing ratios at lower temperatures as illustrated in Table 1. In the experiments investigating SCIs reactions, 90±10 ppb of formic acid (FA, ≥ 98%, Sigma-Aldrich) was evaporated and added before $O_3$ injection. FA was selected as a SCIs scavenger because of its high efficiency of consuming SCIs with a reaction coefficient of larger than $1×10^{-10}$ $cm^3$ $molecule^{-1}$ $s^{-1}$ (Lin and Chao, 2017). $O_3$ was generated by a silent discharge generator (Semozon 030.2, Sorbios) in pure oxygen. $O_3$ concentrations were elevated as temperature decreased due to the positive temperature dependence of the β-pinene ozonolysis reaction, resulting in similar ozonolysis rates at different temperatures. It should be noted that the $O_3$ level used in this chamber study was much higher than the typical ambient level for two reasons: first, for better comparison among different conditions more than 90% of β-pinene was expected to be consumed in each experiment; second, a long residence time was avoided to reduce the impact of wall losses. CO (40% in nitrogen, Basi Schöberl GmbH) was added and used as an OH radical scavenger and a precursor for $HO_2$ radicals. Different CO concentrations were used to modify the $[HO_2]/[RO_2]$ as 0.30–0.34 at low $[HO_2]/[RO_2]$, 1.06–1.26 at middle $[HO_2]/[RO_2]$, and 1.53–1.88 at high $[HO_2]/[RO_2]$. In the following these conditions are denoted as low (L), middle (M), and high (H) $[HO_2]/[RO_2]$ conditions. The simulation results of $[HO_2]/[RO_2]$ are shown in Section 3.1.

Before each experiment, the AIDA chamber was evacuated to around 1 Pa, flushed several times with 10 hPa of synthetic air, and filled to 1 atm with dry or humidified synthetic air. About 1000 $cm^{-3}$ ammonium sulfate (AS) particles (mode diameter: 235–245 nm) were generated by an ultrasonic nebulizer (Sinaptec NA2000), dried, and introduced in each experiment as seed particles to reduce wall losses of the semi- and low-volatile products. In most experiments, to avoid the impact of water vapor on the radical chemistry and the measurements, the water vapor mixing ratio was controlled to be 1–3 ppm. The relative humidity (RH) was increased in the experiments simulating the water vapor interference in the atmosphere. Table 1 shows a summary of the experimental conditions in this study.

### 2.2 Instrumentation

The concentrations of gas-phase β-pinene and lightly oxidized products, such as carbonyls, were measured by a proton-transfer-reaction time-of-flight mass spectrometer (PTR-ToF-MS 4000, Ionicon Analytic GmbH). The data was analyzed by PTR viewer 3.3.12. The inlet flow was 30 standard cubic centimeter per minute (SCCM), and a bypass flow of 3.9 standard

liter per minute (SLM) was added to reduce the residence time in the Silcosteel sampling tube. The PTR-MS was calibrated with a gas standard (Ionicon Analytic GmbH), and a transmission curve was determined to calculate the concentrations of

compounds not present in the gas standard. The sensitivity of β-pinene was $69.7\pm3.6$ cps ppb$^{-1}$ for $10^6$ cps $H_3O^+$. The $O_3$ concentrations were measured by an $O_3$ monitor ($O_3$-41M, Environment). CO concentrations were measured by a CO monitor (NGA 2000, Rosemount Analytic). Water vapor concentrations were measured by a frost point mirror hygrometer (373LX, MBW) and in situ by a tuneable diode laser at 1370 nm (Fahey et al., 2014).

Particle size distributions and number concentrations were measured by a scanning mobility particle sizer (SMPS) consisting

of a differential mobility analyzer (DMA 3071, TSI Inc.) and a condensation particle counter (CPC 3772, TSL Inc.). The sampling flow was 0.3 SLM, and the sheath air flow was 3 SLM. The diameter range measured was 13.6–736.5 nm. A high-resolution time-of-flight aerosol mass spectrometer (HR-ToF-AMS, Aerodyne Inc.) was used to measure the aerosol mass size distribution versus the vacuum aerodynamic diameter. The instrument was calibrated with ammonium nitrate particles. The PIKA v1.80C software was used to analyse the AMS data.

Gas-phase and particle-phase oxidized products were measured by the Filter Inlet for Gas and Aerosols (FIGAERO, Aerodyne Inc.) coupled to a high-resolution time-of-flight chemical ionization mass spectrometer (HR-ToF-CIMS, Aerodyne Inc.) employing $I^-$ for ionisation. Gas-phase samples were collected from the chamber through a 1/4-inch FEP tube at 6 SLM, resulting in a sampling residence time of less than 1 s to reduce the loss in the tube. 2 SLM of the gas flow went to the CIMS and was analyzed online. Particle-phase compounds were collected on PTFE filters (2 μm, SKC Inc.) through a 1/4-inch

stainless-steel tube at 6 SLM. The sampling time of each filter was typically 15–20 minutes adjusted to organic aerosol mass concentrations in the chamber to achieve sufficiently low and similar mass loadings on each filter of about 1 μg.

After collection the filter samples were stored at 253 K. After the experiments these filters were heated by FIGAERO-CIMS using a flow of ultra-high-purity nitrogen as carrier gas following a thermal desorption procedure from 296 K to a maximum temperature of 473 K with a total desorption time of 35 min. Integration of the thermal desorption profiles, i.e., thermograms,

of individual compounds yields their total particle-phase signals. The data were analyzed with the Tofware software v3.1.2, and the reagent ion $I^-$ was subtracted from the mass-to-charge ratio of all the molecules shown below. Background measurements for both the gas and particle phase were done before adding the reactants, and the background signals were subtracted from the results. All ion signals were normalized to $10^6$ cps $I^-$ for comparison, and particle-phase signals were also normalized to the sampling volume. Pinic acid ($C_9H_{14}O_4$), as the most abundant product formed during the reaction, was used

to calibrate the CIMS, and a sensitivity of $12.6\pm1.5$ cps ppt$^{-1}$ for $10^6$ cps $I^-$ was observed. More details about the calibration can be found in the Supplement. In the following, we assumed the same sensitivity for all compounds detected by CIMS and used signal intensity for the comparison. It should be noted that the sensitivity of pinic acid may not represent the sensitivities of all compounds measured by CIMS. Compound responses to $I^-$-CIMS can vary by orders of magnitude, even for similarly structured compounds (Lee et al., 2014). This calibration issue can be problematic if the species distribution of a specified

group changes as a result of the changing experimental conditions. The typical instrumentation and the schematic of the AIDA chamber are shown in Fig. S1.

Since all the instruments and the filter sampling were operated at $295\pm2$ K, the influence of the temperature difference to the simulation chamber needs to be considered (Gao et al., 2023). For the online measurement instruments, a bypass flow was added to reduce the residence time in the sampling tubes to be less than 10 s. Such a short residence time avoids significant

particle evaporation and diminishes artifacts on online measurements. For particles collected on PTFE filters, which were later analyzed by FIGAERO-CIMS, the sampling time of each filter was 15–20 minutes. Due to the short residence time of about 1 s in the sampling line the filter was significantly cooled for low temperature experiments. Before storing the filter samples in a freezer at 253 K there were also 5–10 minutes of handling time. Hence, we cannot rule out that some particulate compounds could evaporate during the sampling, resulting in a potential underestimation of some more volatile compounds in the particle

phase for low temperature experiments. However, considering the dry conditions of our experiments, substantial evaporation of (semi-)volatile compounds from particle phase should be hindered due to the high viscosity of the particles.

## 3 Modeling

### 3.1 Chemical kinetic model

Here we used a box model run by the IDL-based EASY package to simulate the chemical kinetics of the reaction system and
to determine the $[HO_2]/[RO_2]$ selected in the experiments. The $\beta$-pinene reaction mechanism was taken from the Master Chemical Mechanism (MCM) v3.3.1 (http://mcm.york.ac.uk/). Considering the previously reported Criegee intermediates (CIs) formation in $\beta$-pinene ozonolysis and the measurement results from this study, some modifications were applied to $\beta$-pinene-derived CIs in modeling, which was elaborated in Section 4.3. By implementing these updates on CIs, the yield of stabilized $CH_2OO$ decreases from 0.15 to 0.1, and the yield of $C_9$-SCIs increases from 0.1 to 0.32. The OH yield from $\beta$-pinene ozonolysis
decreases slightly from 0.35 to 0.3. It should be noted that for $\beta$-pinene SCIs, only reactions with CO, NO, $NO_2$, $SO_2$, and $H_2O$ are concluded in MCM. When CO was used to adjust $[HO_2]/[RO_2]$ by reacting with OH radicals, the possibility of CO reacting with SCIs needed to be estimated. The reaction coefficients of SCIs with CO are usually reported as smaller than $10^{-18}$ cm$^3$ molecule$^{-1}$ s$^{-1}$ (Eskola et al., 2018; Kumar et al., 2014, 2020; Vereecken et al., 2015), and the reaction coefficient of $10^{-18}$ cm$^3$ molecule$^{-1}$ s$^{-1}$ was applied for SCIs reaction with CO in the current model. For better simulating $C_9$-SCIs, the unimolecular
reaction with a coefficient of 75 s$^{-1}$ and reactions with $HO_2$ and $RO_2$ with coefficients of $2\times10^{-11}$ cm$^3$ molecule$^{-1}$ s$^{-1}$ were implemented in the model, which was elaborated in Section 4.4. These modifications showed a limited impact on $HO_2$ and $RO_2$ concentrations. The modeling results shown below were derived by implementing these updates. The results showed that reaction with CO accounted for less than 1% of SCIs at all temperatures. Another concern of using different CO concentrations was how it would influence the OH reactions in the system. The quantities of $\beta$-pinene consumption by reactions of OH and
$O_3$ were calculated, and the temperature did not significantly impact the oxidation pathways of $\beta$-pinene. At low $[HO_2]/[RO_2]$, the amount of $\beta$-pinene oxidized by OH radicals were only about 1% versus $O_3$ at the end of the experiment, and this ratio increased to about 2.4% and 6%, respectively, at middle and high $[HO_2]/[RO_2]$ conditions as shown in Fig. S3. The results indicate that even at high $[HO_2]/[RO_2]$, ozonolysis was still the dominant reaction pathway in the system and could account for more than 90% of $\beta$-pinene oxidation.

Figure 1 shows the modeled $[HO_2]/[RO_2]$ evolutions for different experimental conditions. With higher $HO_2$ concentration (Fig. S4), $RO_2$ consumption by reacting with $HO_2$ was accelerated, resulting in lower $RO_2$ concentration and higher $[HO_2]/[RO_2]$. The average $[HO_2]/[RO_2]$ before 7000 s of reaction time at low, middle, and high $[HO_2]/[RO_2]$ conditions were calculated to be 0.34, 1.06, and 1.53 at 298 K; 0.33, 1.15, and 1.68 at 273 K; 0.30, 1.26, and 1.88 at 248 K, indicating that $[HO_2]/[RO_2]$ was effectively adjusted during the reaction. In addition to the impact of CO concentration, temperature also has
influence on $[HO_2]/[RO_2]$. The higher $[HO_2]/[RO_2]$ calculated for lower temperatures can be attributed to the negative temperature dependence of the $RO_2+HO_2$ reactions, of which the rate coefficients at 248 K are typically more than twice of those at 298 K. The temperature dependence of $RO_2+RO_2$ reactions was not considered in the box model due to the complexity of this kind of reactions (Atkinson et al., 2006). Here the model sensitivity tests for $RO_2+RO_2$ reactions in $\beta$-pinene oxidation were carried out. For the test of the negative temperature dependence of $RO_2+RO_2$ reactions, the reaction coefficients of
$RO_2+RO_2$ reactions at 273 K and 248 K were modified as 1.5 and 2 times of those at 298 K referring to the variation of $HO_2+RO_2$ reaction coefficient at different temperatures. The results showed that compared to conditions without temperature impact on $RO_2+RO_2$ reactions, the negative temperature dependence led to a decrease of simulated $RO_2$ concentration for about 4% and 10% at 273 K and 248 K. Conversely, when the reaction coefficients of $RO_2+RO_2$ reactions at 273 K and 248 K were modified as 0.75 and 0.5 times of those at 298 K, the positive temperature dependence of $RO_2+RO_2$ caused an increase

for 4% and 10% of simulated $RO_2$ concentrations at 273 K and 248 K. The modeling results showed that changing $RO_2+RO_2$ reaction coefficients within 2 times for different temperature dependences could cause a limited effect on $RO_2$ concentrations.

## 3.2 Aerosol dynamic model

The aerosol dynamic model COSIMA was used to simulate the dynamics of aerosols in the chamber (Naumann, 2003; Saathoff et al., 2009). For products formed from oxidation reactions, gas-particle partitioning and wall loss processes are calculated by 210 this model. As for particles, the coagulation, condensation, evaporation, sedimentation deposition, and diffusion to the walls are calculated. Simulations started with a measured particle size distribution, and an example of the comparison between measured and modeled particle size distribution is shown in Fig. S5.

The AIDA walls can be considered as an irreversible sink especially for acidic gas-phase species and particles, which are important for determining SOA formation. Here the wall loss rates of different species were evaluated. For β-pinene, we 215 observed the time variation of β-pinene before adding $O_3$, and the concentration of β-pinene remained constant for two hours. Two abundant carbonyl products, nopinone and formaldehyde (HCHO), were measured and their concentrations remained constant for two hours, indicating that the wall loss for such kind of carbonyl compounds could be ignored in the timescale of this study. As for $O_3$, it was observed that after almost all β-pinene was reacted, the concentration of $O_3$ kept decreasing. Based on the decreasing tendency, the wall loss rate constants of $O_3$ were estimated to be $(9.0\pm1.0) \times10^{-6}$ s$^{-1}$ at 298 K, $(5.0\pm0.5) \times10^{-6}$ 220 s$^{-1}$ at 273 K, and $(4.0\pm0.5) \times10^{-6}$ s$^{-1}$ at 248 K. Since there is one unsaturated bond in β-pinene molecule, the further $O_3$ oxidation on the products was regarded as negligible. This procedure was supported by experiments where similar ozone wall loss rates were measured in the absence of other compounds. For organic acids, the aluminum wall acts as a significant sink. Here we calculated the wall loss rates of FA and $C_9H_{14}O_4$ (corresponding to the formula of pinic acid and homoterpenylic acid). The wall loss rates of FA were calculated by introducing FA into a clean chamber and observing the decay. As for 225 $C_9H_{14}O_4$, it was generated during the reaction, and the decay rates were calculated when more than 90% of β-pinene was oxidized and the aerosol concentration kept stable. It was estimated that the wall loss rate constant of FA was $(2.5\pm0.5) \times10^{-4}$ s$^{-1}$ at 298 K, $(1.2\pm0.7) \times10^{-4}$ s$^{-1}$ at 273 K, and $(1.1\pm0.3) \times10^{-4}$ s$^{-1}$ at 248 K. For $C_9H_{14}O_4$, the wall loss rate constants were $(2.6\pm0.5) \times10^{-4}$ s$^{-1}$, $(1.2\pm0.4) \times10^{-4}$ s$^{-1}$, and $(1.0\pm0.4) \times10^{-4}$ s$^{-1}$ at 298 K, 273 K, and 248 K, respectively. This is consistent with a limitation of their wall loss rates by diffusion through the laminar layer at the chamber wall. The time series of gas- 230 phase $C_9H_{14}O_4$ before and after wall loss correction are shown in Fig. S6 for different temperatures. These wall loss rates were used in the COSIMA model to simulate the impact of vapor wall losses on SOA formation.

## 4 Results and discussion

### 4.1 Temperature dependence of SOA formation

Figure 2 shows the typical evolution of β-pinene, $O_3$, SOA mass concentrations and mass size distributions at 298 K, and same 235 types of plots for 273 K and 248 K are shown in the Supplement. The line in Fig. 2A represents the SOA mass concentration simulated by the aerosol dynamic model after wall loss correction. With the reaction proceeding, the SOA mass concentration successively increased and reached stable values as a result of both the SOA formation and wall loss processes. In most cases, filter samples were collected after 7000 s, when more than 90% of β-pinene was consumed, ensuring that the quantities of β-pinene oxidized were similar at different experimental conditions. Although seed particles were added as condensational sink, 240 new particle formation still occurred due to the formation of low-volatile products with a strong nucleation capability. The effective density of SOA is determined by comparing the mass and volume size distribution measured by SMPS and AMS, respectively (DeCarlo et al., 2004). The density of SOA formed from β-pinene ozonolysis was calculated as $1.28\pm0.09$ g·cm$^{-3}$, which is in agreement with previously reported values (Bahreini et al., 2005; Kostenidou et al., 2007). The effect of

temperature on SOA density was found to be not significant. More details on SOA density calculation were provided in the Supplement.

The SOA yields are calculated as ratio of the SOA mass concentration ($\mu g \cdot m^{-3}$) versus the mass concentration of β-pinene reacted ($\mu g \cdot m^{-3}$). The SOA yields measured for different experimental conditions are shown in Fig. 3A. The aerosol dynamic model calculations of the particle lost and gases lost to the walls are shown in Fig. 3 for different temperatures. The results demonstrated that the particles lost to the walls accounted for less than 10% of the SOA mass concentrations at all temperatures within the timescale of the experiment. The mass of gases lost to the walls was largest at 298 K of about 25.5% in total SOA mass, and the wall loss effect of gases became smaller with decreasing temperature. The calculated SOA yield at 298 K and low [$HO_2$]/[$RO_2$] was (12.8±1.0) %, and around 16.8% after wall loss correction, which was in the range of previously reported SOA yield for β-pinene ozonolysis (Table S1). When the temperature decreased to 273 K, the SOA yield increased to (15.5±1.1) % (19.0% after wall loss correction) at low [$HO_2$]/[$RO_2$]. With the temperature further decreasing to 248 K, the SOA yield decreased to (9.1±0.6) % (11.6% after wall loss correction).   The impact of temperature on SOA formation in β-pinene ozonolysis was found to be not monotonic. Such a temperature dependence of SOA formation was also observed in β-pinene ozonolysis in von Hessberg et al. (2009) from 263 K to 303 K.

Increasing [$HO_2$]/[$RO_2$] leads to reduced SOA formation at all temperatures, indicating that in β-pinene ozonolysis, $RO_2$+$RO_2$ reactions contribute more to SVOCs and LVOCs formation compared to $RO_2$+$HO_2$ reactions. [$HO_2$]/[$RO_2$] was changed by using different concentrations of CO, resulting in different concentrations of $HO_2$ radicals. Higher $HO_2$ concentrations led to a larger sink for $RO_2$ radicals and consequently a lower $RO_2$ concentration. Additionally, changing [$HO_2$]/[$RO_2$] impacts the branching ratios of product formation from $RO_2$ reactions. Reaction with $HO_2$ is a chain termination for $RO_2$ radicals, leading to the formation of more volatile products, while reactions with $RO_2$ can be chain termination or chain propagation processes. Docherty and Ziemann (2003) proposed that the formation of pinic acid in β-pinene oxidation was inhibited by the increasing [$HO_2$]/[$RO_2$] as the formation pathway of pinic acid involved a series of $RO_2$ chain propagation reactions. The inhibition of increasing [$HO_2$]/[$RO_2$] on pinic acid formation was also observed in this study for all temperatures. The gas-phase composition shown in Fig. S9 demonstrates that most gas-phase products observed in this study are monomers and smaller molecules, while gas-phase dimers only account for a small fraction. Figure 4 shows the volatility distribution of gas-phase products at different [$HO_2$]/[$RO_2$]. Although increasing [$HO_2$]/[$RO_2$] promoted the formation of several compounds, the suppression of increasing [$HO_2$]/[$RO_2$] on gas-phase products observed from CIMS was more obvious. The relative inhibition of increasing [$HO_2$]/[$RO_2$] on gas-phase products was more significant at 273 K and 248 K compared to 298 K. The shift of the volatility classes with decreasing temperature results in the shifts of some gas-phase compounds to lower volatility classes. This could partly explain the larger inhibition effect of increasing [$HO_2$]/[$RO_2$] on SOA yield at lower temperatures.

Compared to low [$HO_2$]/[$RO_2$] condition, the SOA yield decreased by 25–35% for middle [$HO_2$]/[$RO_2$] condition, and 45–70% for high [$HO_2$]/[$RO_2$] condition. The suppression of increasing [$HO_2$]/[$RO_2$] on SOA formation was larger at lower temperatures. Scavenging SCIs inhibited SOA formation substantially at all temperatures, with decreasing SOA yields in the range of 50–70%.  To avoid unwanted particle-phase reactions and interference on the gas-phase CIMS measurement caused by the extremely high FA concentration, we added moderate concentrations of FA. Since the reaction coefficients of β-pinene-derived $C_9$-SCIs are not clear, it is difficult to calculate the proportion of the scavenged SCIs directly. In SCIs scavenging experiments more than 70% of the gas-phase dimers were diminished for all temperatures, based on which we estimated that more than 70% of $C_9$-SCIs were scavenged. The results showed the importance of SCIs and $RO_2$ in SOA formation from 248 K to 298 K, and their reaction mechanisms and products formation deserved further analysis.

## 4.2 SOA composition and abundance of dimers

Monomers with molecule formulas of $C_{8–10}H_{8–20}O_{3–10}$ and dimers with molecule formulas of $C_{16–20}H_{22–40}O_{4–12}$ were identified by FIGAERO-CIMS in the particle phase. The normalized signals of all particulate $C_xH_yO_z$ are shown in Fig. S10. Monomers

and dimers usually account for 54–64% and 12–20% of total particle-phase signal intensities. Fractions of different dimer species are shown in Fig. 5A. The $C_{20}$ dimers had a lower abundance accounting for only ~ 5% of all dimers. This can be explained by the fact that the major SCIs and $RO_2$ formed from β-pinene ozonolysis contain 8 or 9 carbon atoms. The fractions of $C_{17}$, $C_{18}$, and $C_{19}$ dimers were 30–40%, 20–40%, and 10–20% in total dimers. For $C_{16}$ dimers, their fraction in dimers decreased from 25–40% at 298 K to ~10% at 248 K. Since we could not calibrate the sensitivity of our CIMS for dimers, assuming a similar sensitivity as that of pinic acid, the contribution of dimers to the SOA mass was estimated to be 17%–21%. As for monomers, the $C_8$ and $C_9$ products were major contributors, which accounted for about 55% and 35% of monomers. $C_{10}$ products usually accounted for less than 10% of total particulate monomers because $C_{10}$ monomers mainly formed from β-pinene reaction with OH. The fractions of $C_{8-10}$ monomers in total particulate composition were similar under different temperature conditions.

The impact of $[HO_2]/[RO_2]$ on gas-phase dimers is shown in Fig. S11, demonstrating the dimers formed from $RO_2+RO_2$ reactions in the gas phase were inhibited with increasing $[HO_2]/[RO_2]$. The relative inhibition was more significant at 273 K and 248 K than at 298 K. Scavenging SCIs resulted in significant suppression on gas-phase dimers for all temperatures as shown in Fig. S12, indicating that dimers formation from gas-phase SCIs reactions were hindered. Since the volatilities of these dimers are presumably sufficiently low enough, they should be primarily in the particle phase even at 298 K, which is in agreement with the FIGAREO-CIMS measurement in both phases. The changing of gas-phase formation pathways of dimers also had a significant impact on particulate dimers, which was shown in Fig. S13. The linear correlation between monomers and dimers in the gas phase shown in Fig. S14 indicates that the dimers identified here are not formed from the clustering of closed-shell monomers. Similar observations were reported for α-pinene oxidation (Zhao et al., 2018). Some particle-phase reactions are reported to influence the formation and decomposition of dimers (Pospisilova et al., 2020; Renbaum-Wolff et al., 2013). Most of our experiments were conducted at dry conditions, leading to a relatively high viscosity of the aerosols and slow particle-phase diffusion. Hence, the potential impact of particle-phase reactions on dimers was limited. We also observed a linear correlation between the particle-phase dimers and monomers as shown in Fig. S15, which suggested that the contribution of particle-phase clustering of monomers to the dimers observed in this study was limited. Although the contribution of particle-phase reactions on dimers cannot be excluded completely, our observations can be explained well by gas-phase reactions.

Figure 5B shows the temperature dependence of the relative abundance of $C_{16-19}$ dimers at low $[HO_2]/[RO_2]$, indicating that differences exist in their formation mechanisms. For $C_{16}$ dimers, their abundances at 298 K and 273 K were similar, while they decreased by more than 60% at 248 K, suggesting that the major formation pathway of $C_{16}$ dimers was inhibited to a large extent. For $C_{17-19}$ dimers, their formation showed increase of about 40% when temperature decreased from 298 K to 273 K. When the temperature further decreased to 248 K, $C_{17}$ dimers decreased by about 45%, and $C_{18}$ and $C_{19}$ dimers decreased by about 35%, indicating the influence of lower temperature on the formation pathways also happened to $C_{17-19}$ dimers. The gas-phase dimers account for less than 5 % of the total gas- and particle-phase dimer signals, which means that most dimers have sufficiently low vapor pressures and primarily stay in the particle phase even at 298 K. Due to their low gas phase concentrations potential wall losses via the gas phase can have only a small impact. Besides, the wall loss of particle mass was calculated to be usually below 10%, suggesting the wall loss effect on the particulate dimers can be regarded as of limited importance. Therefore, changes of these particulate dimers with temperature can be mainly attributed to the impact of temperature on their formation pathways. One possible reason for the inhibition of dimers' formation at 248 K is the temperature impact on HOMs formation, since the autoxidation rate coefficients are prompted rapidly by increasing temperature (Praske et al., 2017). Ehn et al. (2014) reported that the formation of HOMs, which could also be regarded as extremely low-volatile organic compounds (ELVOCs), was about two orders of magnitude lower in β-pinene ozonolysis than in α-pinene ozonolysis. In this study, the HOMs observed (monomers with 6–10 and dimers with 8–12 oxygen atoms) were a small part of the total particle-phase monomers and dimers' signal, indicating that the decreasing autoxidation rate at lower

temperatures could not fully explain the suppression of dimers formation below 273 K. In the next section, we will discuss the contribution of other formation pathways to dimers.

It is noted that some monomers show two peaks in their thermograms. Figure S16 shows the thermograms of two abundant monomers formed during the reaction, i.e., $C_8H_{12}O_4$ (corresponding to the formula of terpenylic acid) and $C_9H_{14}O_4$ (corresponding to the formula of pinic acid and homoterpenylic acid). Lopez-Hilfike et al. (2015) measured similar thermograms of $C_8H_{12}O_4$ and $C_9H_{14}O_4$ from aerosols generated in α-pinene oxidation, and they claimed that according to their calibrated desorption-temperature relation, it was unlikely that such a big difference existed between the vapor pressures of isomers, suggesting that the thermal decomposition of some unstable oligomers, which probably contained noncovalent bonds, contributed to the second desorption peak. We estimated that the fraction of the second peak in the thermograms of $C_8H_{12}O_4$ and $C_9H_{14}O_4$ accounted for 30–50%. The thermograms of two abundant dimers $C_{17}H_{26}O_8$ and $C_{18}H_{28}O_6$ are shown in the figure for comparison. These two dimers showed one peak in their thermograms, in which the desorption temperatures corresponding to the peak signal ($T_{max}$) were around 100 °C. Although a small fraction of the dimers showed double peaks in their thermograms, the fraction of the second peak in the total signal was usually less than 35%. Due to the thermal decomposition of some unstable dimers, the signal fraction of dimers in $C_xH_yO_z$ reported in this study represents a lower limit. The sum thermograms at different temperatures are shown in Fig. S17, and the $T_{max}$ of all the monomers and dimers are summarized in Table S2.

## 4.3 Chemistry of SCIs and their impact on dimers

The scavenging of SCIs leads to a reduction of more than 40% in total dimers from 248 K to 298 K (Fig. 6), indicating the significant contribution of β-pinene-derived SCIs to dimers formation. After the addition of $O_3$ to β-pinene, the primary ozonide generates and usually has two decomposition pathways. One leads to excited $C_9$-CIs and HCHO, and the other forms excited $CH_2OO$ and nopinone. In previous studies, the formation of excited $C_9$-CIs was reported to be the primary pathway, which could account for 80–90% of the primary ozonide decomposition (Ma and Marston, 2008; Nguyen et al., 2009). Figure S18 shows the formation of HCHO and nopinone as a function of reacted β-pinene at different [$HO_2$]/[$RO_2$] for all temperatures. Both the formation of HCHO and nopinone show good linear correlation with β-pinene reacted. Different [$HO_2$]/[$RO_2$] caused by different CO concentrations did not influence HCHO and nopinone formation, confirming that the CO reaction with SCIs was negligible in this study. No obvious temperature impact on HCHO and nopinone formation was observed, indicating that temperature did not influence the early reaction steps of CIs' generation. The molar yields of HCHO and nopinone are calculated as 0.63±0.06 and 0.16±0.02, which are in the range of values reported previously in Table S3(Elayan et al., 2019; Lee et al., 2006; Winterhalter et al., 2000). Considering the yields of HCHO and nopinone observed in this study and the values suggested in previous studies, the branching ratios of the formation of the excited $C_9$-CIs and the excited $CH_2OO$ from ozonide decomposition were modified from 0.6 and 0.4 to 0.8 and 0.2 in the MCM mechanism. It was reported that the yield of $C_9$-SCIs was about 0.35 in β-pinene ozonolysis, and the yield of stabilized $CH_2OO$ was about 0.1 (Ahrens et al., 2014; Winterhalter et al., 2000; Zhang and Zhang, 2005). Based on this the branching ratios of forming $C_9$-SCIs and stabilized $CH_2OO$ from excited Criegee intermediates (ECIs) were adjusted to be 0.4 and 0.5. Another important reaction pathway of ECIs is isomerization and decomposition, forming OH radicals. The OH yield from β-pinene ozonolysis was reported to be about 0.3, which is half of that from α-pinene ozonolysis (Atkinson et al., 1992; Nguyen et al., 2009). Table S4 shows the summary of the main updates to the formation of β-pinene-derived CIs in the MCM mechanism.

Scavenging of SCIs led to a significant suppression of dimers and SOA formation for different [$HO_2$]/[$RO_2$] conditions as shown by Fig. 6. The addition of SCIs scavenger was reported to lead to a decrease of $RO_2$ concentrations in the range of 11−17%, which was not critical for the results (Berndt et al., 2018b). At all temperatures, more than 50% of SOA formation was inhibited by scavenging SCIs, while dimers showed different temperature dependences. For $C_{18}$ and $C_{19}$ dimers, their decreases by scavenging SCIs were more than 50% at all temperatures, indicating that SCIs reactions are always a dominant

source for $C_{18}$ and $C_{19}$ dimers. For $C_{16}$ and $C_{17}$ dimers, and especially for $C_{16}$ dimers, the impact of scavenging SCIs varied with temperatures. At 298 K, the scavenging of SCIs showed a limited impact on $C_{16}$ dimers' generation with about 20% decrease. With the temperature decreasing, the relative contribution of SCIs reactions for $C_{16}$ dimers formation became larger. Considering the inhibited formation of $C_{16}$ dimers at lower temperatures, it was attributed that the main formation pathway of

$C_{16}$ dimers was largely limited at low temperatures, resulting in an increase of the relative importance of SCIs reactions in $C_{16}$ dimers formation.

To further investigate the dimers formation mechanism through SCIs reaction channel at different temperatures, the contribution of SCIs to individual dimers was paid attention to. The most abundant dimers $C_{16-19}H_{22-32}O_{5-9}$ ($C_{16}H_{22-28}O_{6-9}$, $C_{17}H_{22-30}O_{5-9}$, $C_{18}H_{24-30}O_{4-8}$, $C_{19}H_{26-32}O_{5-8}$), which accounted for more than 70% of total dimer signals, were selected for

further analysis of their formation mechanisms. For this purpose, we defined that if one dimer was suppressed by $\geq$ 50% when scavenging SCIs at all temperatures, it was classified as a SCIs-controlled dimer. For these selective abundant dimers, most of $C_{18}$ and $C_{19}$ dimers and half of $C_{17}$ dimers are SCIs-controlled, while none of the $C_{16}$ dimers are mainly contributed by SCIs reactions. $C_9$-SCIs contributed to $C_{17-19}$ dimers through reacting with $C_{8-10}$ products, which were more abundant in the gas phase compared to $C_7$ products. Figure 7 shows the relative changes of SCIs-controlled and non-SCIs-controlled abundant

dimers at 298 K or 248 K versus 273 K. For the dimers mainly controlled by SCIs reactions, more than half of them showed higher abundances at 248 K than at 298 K, suggesting that the contribution of SCIs reactions to these dimers was not suppressed, even though the gas-phase oxidized monomers' concentrations were lowest at 248 K. For non-SCIs-controlled dimers, they usually had much higher formation at 298 K than 248 K. The results demonstrate the importance of SCIs in contributing to dimers and SOA formation at lower temperatures.

When considering the SOA formation potential of SCIs in the atmosphere, one limiting factor is the water vapor concentration. Although the reaction coefficient between SCIs and $H_2O$ is not fast, and this reaction depends on the structure of SCIs, it is still one of the most important sinks for SCIs due to the high concentrations of water vapor in the atmosphere (Lin and Chao, 2017). Through raising RH to 15% at 298 K ($H_2O$: $1.15\times10^{17}$ molecule $cm^{-3}$) and 80% at 273 K ($H_2O$: $1.29\times10^{17}$ molecule $cm^{-3}$), about 40% of inhibition on dimers' formation was observed. When increasing RH to 70% at 248 K ($H_2O$: $1.73\times10^{16}$

molecule $cm^{-3}$), there was no obvious suppression on dimers, suggesting that the contribution of $\beta$-pinene to atmospheric SCIs and dimers could be more important in colder regions because of the lower water vapor concentration. The water vapor concentration could also influence the peroxy radical chemistry, while here this issue was not analyzed in detail.

## 4.4 Specific $[HO_2]/[RO_2]$ impact at lower temperatures

The changing $[HO_2]/[RO_2]$ showed significant impact on dimers formation, especially for lower temperatures, suggesting the

influence of lower temperatures on $RO_2$ reactions. The relative changes of particulate dimers and SOA with increasing $[HO_2]/[RO_2]$ are shown in Fig. 8, which illustrates that the formation of dimers becomes more sensitive to $[HO_2]/[RO_2]$ changes at lower temperatures. At 298 K, the decrease of dimers was within 10% from middle $[HO_2]/[RO_2]$ to high $[HO_2]/[RO_2]$, and this value increased to about 20 % and 30% at 273 K and 248 K, respectively. At high $[HO_2]/[RO_2]$, the dimers abundances were about 69%, 56%, and 30% of those at low $[HO_2]/[RO_2]$ at 298 K, 273 K, and 248 K, respectively. $[HO_2]/[RO_2]$ impacted

the gas-phase dimers formation from $RO_2$+$RO_2$ reactions as follows:

$$[ROOR] = \gamma \cdot [RO_2]^2 \qquad (1)$$

Where $[RO_2]$ is the concentration of $RO_2$ in the gas phase; $[ROOR]$ is the concentration of dimers formed in the gas phase and quickly partitioned to the particle phase; $\gamma$ is the branching ratio of dimers formation from $RO_2$+$RO_2$ reactions. The $RO_2$ concentrations were simulated in the box model for different conditions and are shown in Fig. S19. If $RO_2$ radicals influence

dimers formation predominately through $RO_2$+$RO_2$ reactions, which are second-order reactions and should therefore lead to a linear correlation of dimer signal with $[RO_2]^2$. Zhao et al. (2018) showed a quadratic relationship between the gas-phase signals of dimers and $RO_2$ as evidence of dimers formation from $RO_2$+$RO_2$ reactions in $\alpha$-pinene ozonolysis. In this study, we would

like to show different impacts of $[HO_2]/[RO_2]$ on dimers formation at different temperatures. Figure 9 shows the correlations between dimers in both phases and simulated $[RO_2]^2$ at different temperatures. At 298 K, the dimer signals show a nearly linear relationship with $[RO_2]^2$. When $[RO_2]^2$ decreased by about 85%, dimers formation was inhibited for around 30%, indicating that the contribution of other pathways to dimers was also important in β-pinene ozonolysis. The correlations between dimers and $[RO_2]^2$ at 273 K and 248 K became different from that at 298 K, suggesting that at lower temperatures there were different impacts of $RO_2$ on dimer formation pathways. Below we will discuss the possible reasons for the specific impact $[HO_2]/[RO_2]$ on dimers at lower temperatures.

It is intriguing to find that at low temperatures, the variation of $[HO_2]/[RO_2]$ has such a big influence on $C_{18}$ dimers (Fig. 8), which are significantly contributed by SCIs reactions. The particular impact of $[HO_2]/[RO_2]$ at low temperatures on $C_{18}$ dimers could be clearly represented by the formation of $C_{18}H_{28}O_6$, one of the most abundant dimers mainly generated from $C_9$-SCIs reaction with $C_9H_{14}O_4$. The gas-phase concentration of $C_{18}H_{28}O_6$ decreases substantially if scavenging SCIs as Fig. 9, confirming that SCIs reactions are the dominant source for $C_{18}H_{28}O_6$. Figure 10A shows the total abundance of $C_{18}H_{28}O_6$ in the gas and particle phase, and the FIGAERO-CIMS measurements showed that more than 90% of them stayed in the particle phase for all temperatures. At 298 K the formation of $C_{18}H_{28}O_6$ was not sensitive to changing $[HO_2]/[RO_2]$, while with temperature decreasing to 273 K and 248 K, the $[HO_2]/[RO_2]$ had an enlarging impact. We evaluated the influence of $C_9$-SCIs reactions with CO at lower temperatures by comparing the formation of nopinone. The formation of nopinone, as the main product from SCIs reaction with CO, is not notably influenced by changing temperatures (Fig. S20), confirming that the CO consumption on $C_9$-SCIs is negligible. $C_9$-SCIs contribute to the formation of $C_{18}H_{28}O_6$ mainly through reacting with $C_9H_{14}O_4$ in the gas phase, and the gas-phase concentrations of $C_9H_{14}O_4$ at different $[HO_2]/[RO_2]$ needed to be compared. Figure S21 shows the gas-phase $C_9H_{14}O_4$ concentrations after wall loss correction at different temperatures, demonstrating the limited effect of $[HO_2]/[RO_2]$ on $C_9H_{14}O_4$ formation. After excluding the possible influence of CO and $C_9H_{14}O_4$, there is still the option that the $[HO_2]/[RO_2]$ directly impacts SCIs reactions with $RO_2$ radicals, which contributes to dimers formation (Chhantyal-Pun et al., 2020b; Sakamoto et al., 2017).

The potential contribution of $C_9$-SCIs reaction with $RO_2$ radicals to dimers formation was evaluated by modeling. Simulations at 298 K were regarded as the basic scenario using the model described in Section 3. Simulations at 248 K were chosen for comparison by implementing the proposed reaction coefficients described below. During the reaction, the unimolecular reaction of SCIs, including isomerization and decomposition, was the main sink of SCIs and was crucial for determining the lifetime of SCIs (Cox et al., 2020). Although some studies reported the unimolecular reaction coefficients of simple SCIs, less is known about the unimolecular reactions of monoterpene-derived SCIs. Gong et al. (2021) estimated the unimolecular reaction coefficient of limonene-derived SCIs as 30 s$^{-1}$ and 100 s$^{-1}$ for different SCIs isomers at 298 K. According to this, we assumed that the unimolecular reaction coefficient of β-pinene-derived $C_9$-SCIs was 75 s$^{-1}$ at 298 K in modeling. The rate coefficients of SCIs unimolecular reactions are strongly influenced by temperature. It was reported that the unimolecular reaction coefficient of $(CH_3)_2COO$-SCIs increased by a factor of four with temperature increasing by 40 K (Smith et al., 2016). As for $CH_3CHOO$-SCIs, the unimolecular reaction coefficient increased by a factor of five with temperature increasing by 35 K (Robinson et al., 2022). Berndt et al. (2014) reported that the ratio of the $(CH_3)_2COO$-SCIs unimolecular reaction coefficient versus the reaction coefficient with $SO_2$ increased by a factor of 34 from 278 K to 343 K. Based on these temperature dependencies, the unimolecular reaction coefficient of $C_9$-SCIs was assumed to decrease by a factor of 5 from 298 K to 248 K. Chhantyal-Pun et al. (2020b) claimed that the reaction coefficient of $CH_2OO$ with $RO_2$ was $(2.4\pm1.2) \times 10^{-11}$ cm$^3$ molecule$^{-1}$ s$^{-1}$. Zhao et al. (2017) reported a negative temperature dependence of substituted alkyl peroxy radicals' reaction with CIs, of which the reaction coefficient would increase one order of magnitude for temperature decreasing from 400 K to 250 K. Based on the reported values, the reaction coefficient of SCIs and $RO_2$ was set as $2\times10^{-11}$ cm$^3$ molecule$^{-1}$ s$^{-1}$ at 298 K, and $8\times10^{-11}$ cm$^3$ molecule$^{-1}$ s$^{-1}$ at 248 K. The radicals formed from SCIs reaction with $RO_2$ further react with $HO_2$ and $RO_2$, generating closed-shell dimers. The model results of dimers formation from $C_9$-SCIs reaction with $RO_2$ are shown in Fig. 10. The modeled

dimers formed from $C_9$-SCIs reaction with $RO_2$ could account for less than 5% in total measured dimers at 298 K, and this value increased to more than 60% at 248 K, indicating a greater contribution of this reaction channel to dimers formation at 248 K. Compared to low $[HO_2]/[RO_2]$ condition, the modeled dimers formed from $C_9$-SCIs reaction with $RO_2$ decreased by 44% and 61% at middle and high $[HO_2]/[RO_2]$ conditions, which helped to explain the observed $[HO_2]/[RO_2]$ influence on dimers at lower temperatures. The higher stability of SCIs at lower temperatures also promoted the bimolecular reactions of SCIs with other closed-shell products, however, due to the decrease of gas-phase concentrations of those closed-shell products, and the potential temperature effect on dimers formation from these reactions, in this study the contribution of this reaction channel seemed to be more important at 298 K than 248 K. It is claimed that the temperature impacts on the rate coefficients and product branching ratios of monoterpene-derived SCIs reactions are still not well defined and need further study.

To better transfer the findings in this study to the real atmosphere, box model simulations at ambient $O_3$ level of 40 ppb, and β-pinene of 2 ppb were conducted at 298 K and 248 K. The modeling for more atmospheric conditions focused on $O_3$ reaction, and the OH concentration was set at a low level ($1 \times 10^4$ molecule cm$^{-3}$). Different $HO_2$ concentrations of $2 \times 10^7$ molecule cm$^{-3}$, $1 \times 10^8$ molecule cm$^{-3}$, and $5 \times 10^8$ molecule cm$^{-3}$ were used for deriving different $[HO_2]/[RO_2]$ as 0.05–0.1, 0.8–2.5, and >20. Similar mechanisms elaborated in Section 3 were utilized and the proposed reaction coefficients of SCIs were utilized at 248 K. The simulations lasted for 7 days, leading to that the accumulated oxidized β-pinene was 23.5 ppb at 298 K and 12.7 ppb at 248 K. The results showed that with $[HO_2]/[RO_2]$ of 0.05–0.1, accumulated concentrations of dimers formed from $C_9$-SCIs with $RO_2$ were $2.26 \times 10^7$ molecule cm$^{-3}$ at 298 K and $1.18 \times 10^8$ molecule cm$^{-3}$ at 248 K. With $[HO_2]/[RO_2]$ increasing to 0.8–2.5 and >20, dimers formed from this channel decreased to $6.57 \times 10^6$ molecule cm$^{-3}$ and $1.28 \times 10^6$ molecule cm$^{-3}$ at 298 K, and $2.44 \times 10^7$ molecule cm$^{-3}$ and $4.28 \times 10^6$ molecule cm$^{-3}$ at 248 K. The atmospheric relevant simulations demonstrated that the variation of temperature and $[HO_2]/[RO_2]$ can have a significant influence on the dimers formation also for ambient $O_3$ levels. It should be noted that in the current simulation, only β-pinene-derived $RO_2$ were considered, while in the real atmosphere, the $C_9$-SCIs have opportunities to react with $RO_2$ formed from other VOCs.

## 5 Conclusions

This study reveals the role of $RO_2$ radicals and SCIs in the formation of dimers and SOA in β-pinene ozonolysis at different temperatures, especially for colder conditions. Both of the reactive intermediates showed their significant influence on SOA yield and composition. Temperature not only impacts the compounds' volatilities, but also impacts the reaction mechanisms and products formation of $RO_2$ and SCIs reactions. The SOA yield is not monotonic with decreasing temperature in β-pinene ozonolysis due to the joint influence of volatilities and chemical mechanisms. Such influence with varying temperatures may exist in other VOCs oxidation systems with higher SCIs yields contributing to SOA mass formation. This finding may help to explain the controversy on the temperature dependence of SOA yield and composition. The SOA formation potential of β-pinene is influenced by several parameters in the atmosphere, such as temperature, RH, and $[HO_2]/[RO_2]$, which are to a large extent correlated to the chemistry of CIs and peroxy radicals. Therefore, these parameters need to be accounted for to represent the SOA formation potential of β-pinene in atmospheric models. The results provide evidence for the importance of SCIs in dimers and SOA formation at subzero temperatures, and the reactions of SCIs and $RO_2$ with negative temperature dependence show increasing importance with decreasing temperature. These results can be used to improve the chemical mechanism modeling of monoterpenes and also SOA parameterization in transport models. The lifetime of SCIs becomes longer in colder regions due to lower temperatures and lower water vapor concentrations, while these intermediates still maintain high reactivities, suggesting the chemistry of SCIs plays an important role and requires more attention in winter and at higher altitudes.

**Data availability.** The data are available via the repository KITopen (link to be added). Data are also available upon request to the corresponding author.


**Supplement.** The supplement related to this article is available online.

**Author contributions.** YG and HS designed the study and carried out the experiments. FJ helped operate the instruments. YL operated PTR-MS and analyzed its data. YG analyzed the data, ran the models, and formatted the manuscript. All co-authors

commented on the manuscript.

**Competing interests.** At least one of the co-authors is a member of the editorial board of Atmospheric Chemistry and Physics. The authors have no other competing interests to declare.

**Acknowledgments.** Technical support by the AIDA staff at IMK-AAF is gratefully acknowledged. YG is grateful to the Helmholtz-OCPC Fellowship Program.

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

**Table 1. Summary of experimental conditions.**

| Exp. | β-pinene (ppb) | O$_3$ (ppm) | CO (ppm)[*] | Formic acid (ppb) | T (K) | RH (%) | [HO$_2$]/[RO$_2$] |
|------|------|------|------|------|------|------|------|
| 298a | 19.3±1.2 | 0.93±0.05 | 25 | 0 | 298±0.3 | < 0.1 | L (low) |
| 298b | 19.3±1.2 | 0.93±0.05 | 100 | 0 | 298±0.3 | < 0.1 | M (middle) |
| 298c | 19.3±1.2 | 0.93±0.05 | 400 | 0 | 298±0.3 | < 0.1 | H (high) |
| 298d | 19.3±1.2 | 0.93±0.05 | 25 | 90±10 | 298±0.3 | < 0.1 | L |
| 298e | 19.3±1.2 | 0.93±0.05 | 100 | 90±10 | 298±0.3 | < 0.1 | M |
| 298f | 19.3±1.2 | 0.93±0.05 | 25 | 0 | 298±0.3 | 14.7±1.2 | L |
| 273a | 18.2±1.0 | 1.10±0.05 | 23 | 0 | 273±0.3 | < 0.1 | L |
| 273b | 18.2±1.0 | 1.10±0.05 | 92 | 0 | 273±0.3 | < 0.1 | M |
| 273c | 18.2±1.0 | 1.10±0.05 | 366 | 0 | 273±0.3 | < 0.1 | H |
| 273d | 18.2±1.0 | 1.10±0.05 | 23 | 90±10 | 273±0.3 | < 0.1 | L |
| 273e | 18.2±1.0 | 1.10±0.05 | 92 | 90±10 | 273±0.3 | < 0.1 | M |
| 273f | 18.2±1.0 | 1.10±0.05 | 23 | 0 | 273±0.3 | 81.3±1.0 | L |
| 248a | 16.3±0.8 | 1.30±0.05 | 21 | 0 | 248±0.3 | < 0.1 | L |
| 248b | 16.3±0.8 | 1.30±0.05 | 84 | 0 | 248±0.3 | < 0.1 | M |
| 248c | 16.3±0.8 | 1.30±0.05 | 333 | 0 | 248±0.3 | < 0.1 | H |
| 248d | 16.3±0.8 | 1.30±0.05 | 21 | 90±10 | 248±0.3 | < 0.1 | L |
| 248e | 16.3±0.8 | 1.30±0.05 | 84 | 90±10 | 248±0.3 | < 0.1 | M |
| 248f | 16.3±0.8 | 1.30±0.05 | 21 | 0 | 248±0.3 | 70.5±1.8 | L |

[*] Uncertainty of 5%.

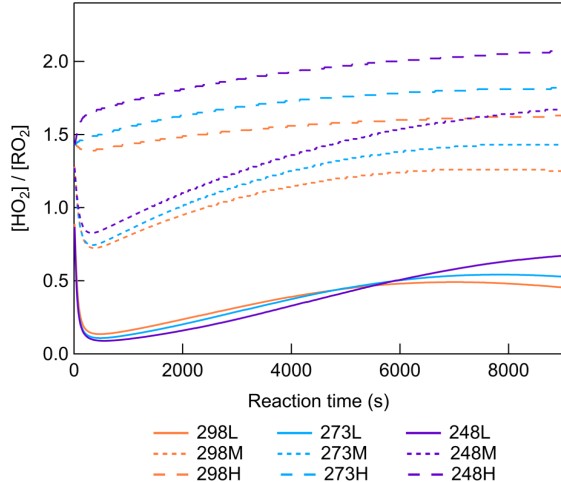

**Figure 1. Simulated [HO₂]/[RO₂] as a function of reaction time at different [HO₂]/[RO₂] conditions and different temperatures (Exp. 298abc, 273abc, 248abc).**

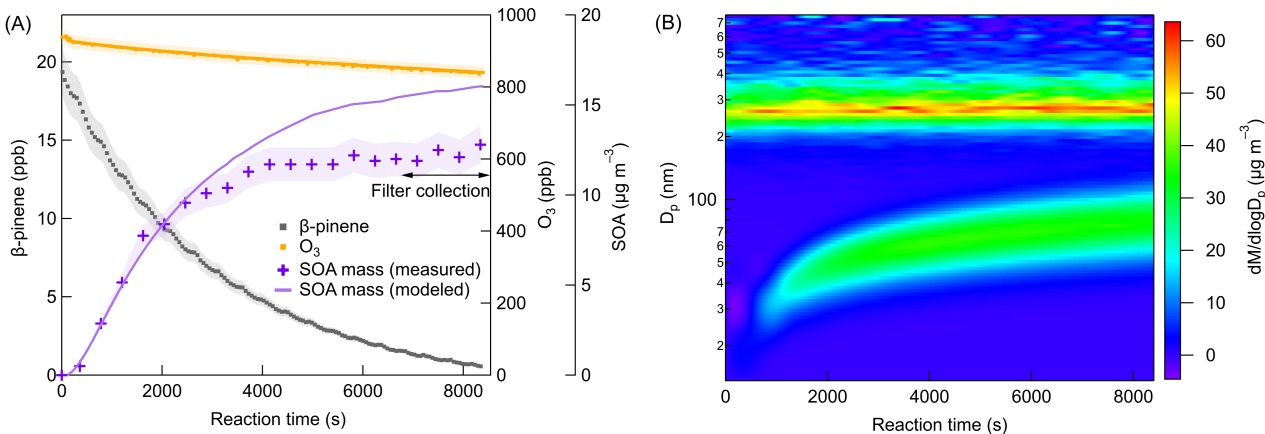

**Figure 2. Time series of (A) β-pinene mixing ratio, O₃ mixing ratio, measured SOA mass concentrations, and simulated SOA mass concentrations after wall loss correction (B) Particle mass size distributions (dM/dlogD_p) at 298 K (Exp. 298a).**

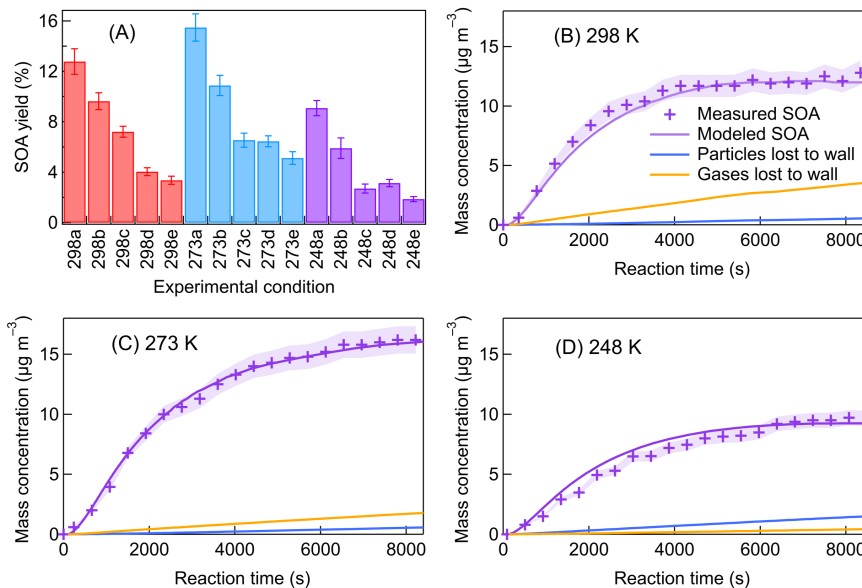

Figure 3. (A) SOA yields at different experimental conditions (a−c: increasing [HO$_2$]/[RO$_2$]; d, e: scavenging SCIs). Measured and modeled SOA mass concentrations, and the wall losses of particles and gases at (B) 298 K (C) 273 K (D) 248 K (Exp. 298a, 273a, 248a).

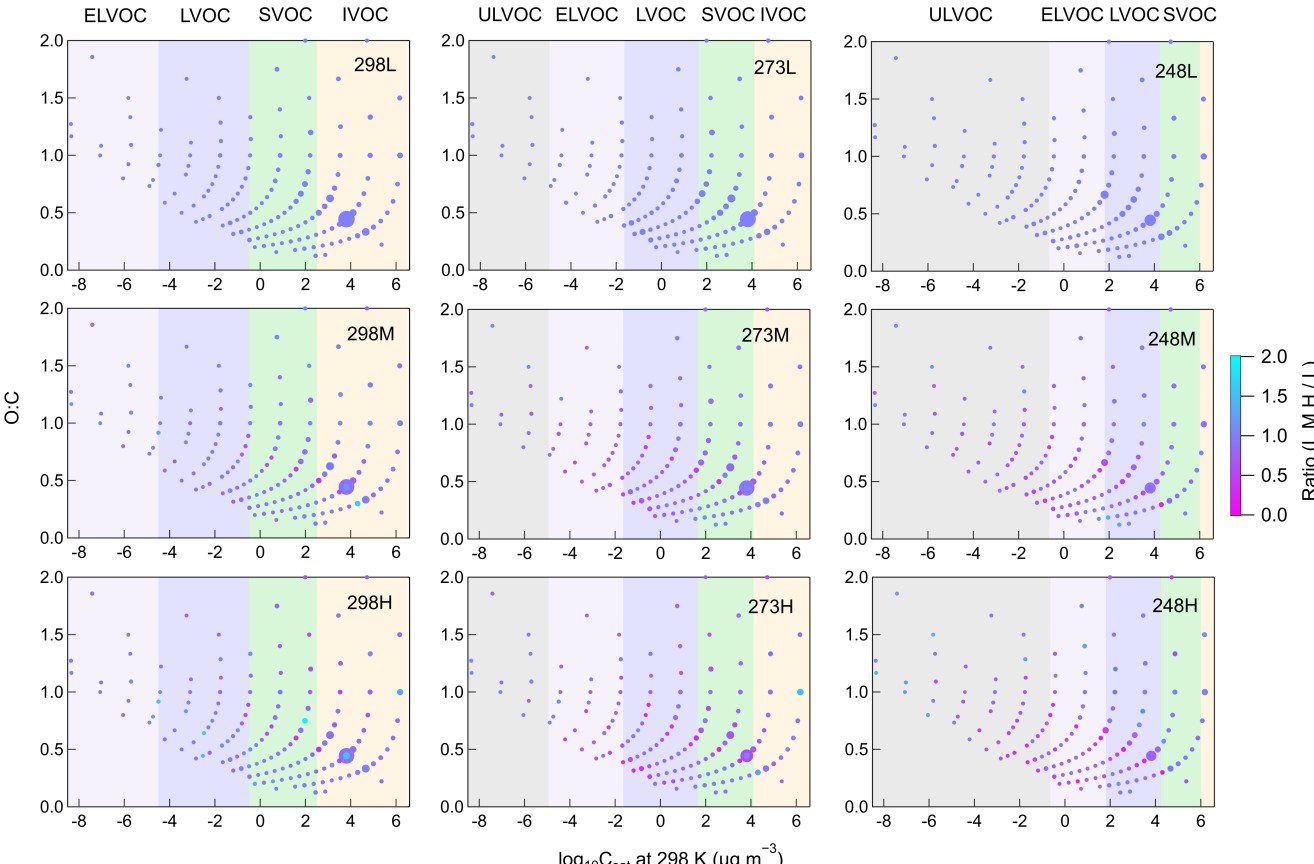

**Figure 4. Volatility distribution of gas-phase products for different temperatures and [HO₂]/[RO₂] (Exp. 298abc, 273abc, 248abc).**
Markers are sized by the square root of their relative abundance and colored by the ratio of signals at different [HO₂]/[RO₂] versus signals at low [HO₂]/[RO₂] at each temperature. From left to right, colored bands in the background represent the volatility classes of ULVOC, ELVOC, LVOC, SVOC, and IVOC. These volatility classes are defined for 298 K and shift with temperature according to the Clausius-Clapeyron equation.

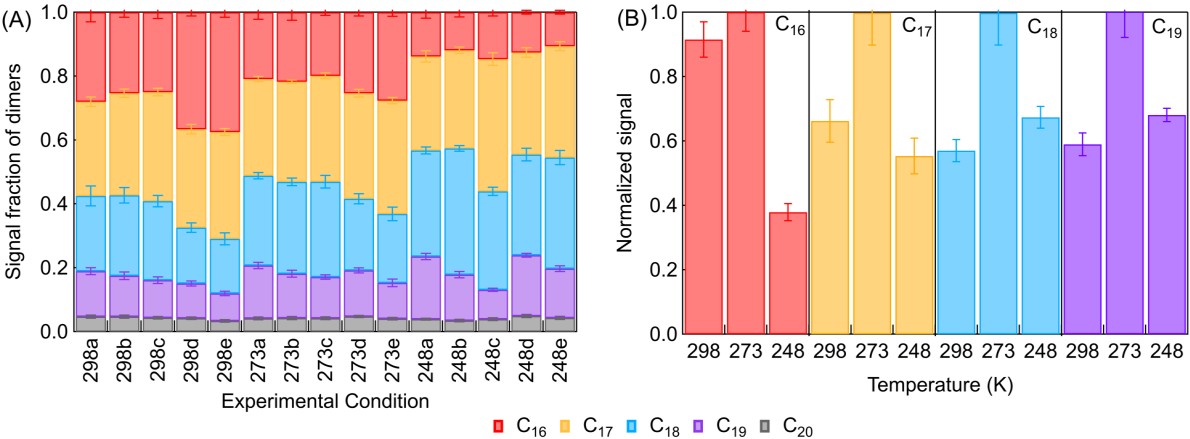

**Figure 5. (A)** Fractions of different dimer species of all particle-phase dimers (a−c: increasing [HO$_2$]/[RO$_2$]; d, e: scavenging SCIs). **(B)** Temperature dependence of the relative abundance of different dimers (Exp. 298a, 273a, 248a).

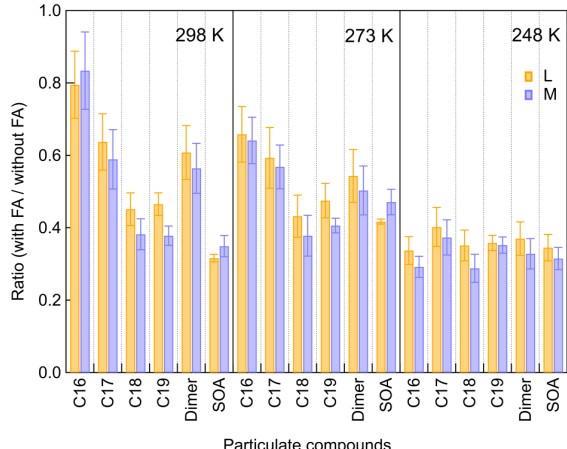

**Figure 6. The relative changes of particulate dimers and SOA yields after scavenging SCIs at low (L) and middle (M) [HO₂]/[RO₂] (Exp. 298de, 273de, 248de).**

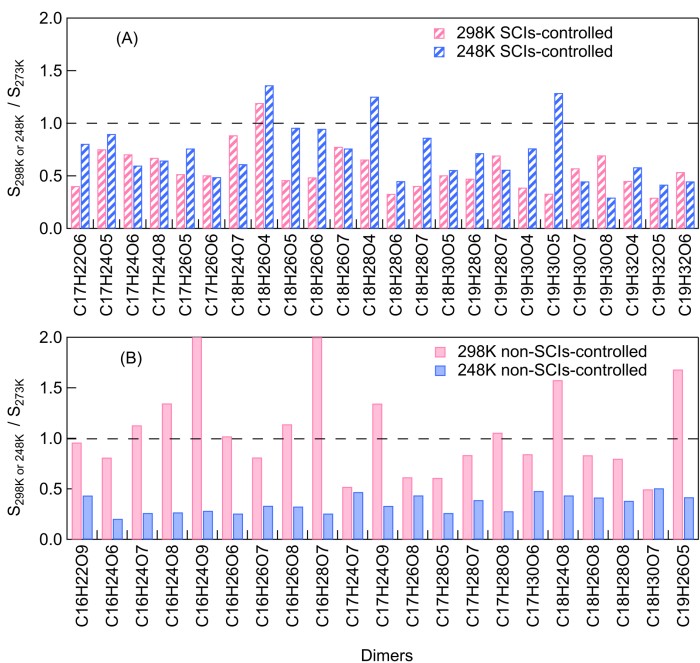

**Figure 7.** The relative changes of (A) SCIs-controlled and (B) non-SCIs-controlled abundant dimers at 298 K or 248 K versus 273 K (The relative standard deviations are within 25 %, Exp. 298a, 273a, 248a).

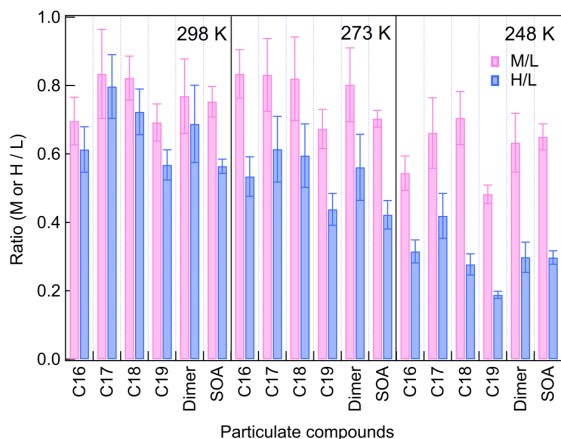

**Figure 8. The relative changes of particulate dimers and SOA yields at middle (M) and high (H) [HO$_2$]/[RO$_2$] compared to low (L) [HO$_2$]/[RO$_2$] (Exp. 298abc, 273abc, 248abc).**

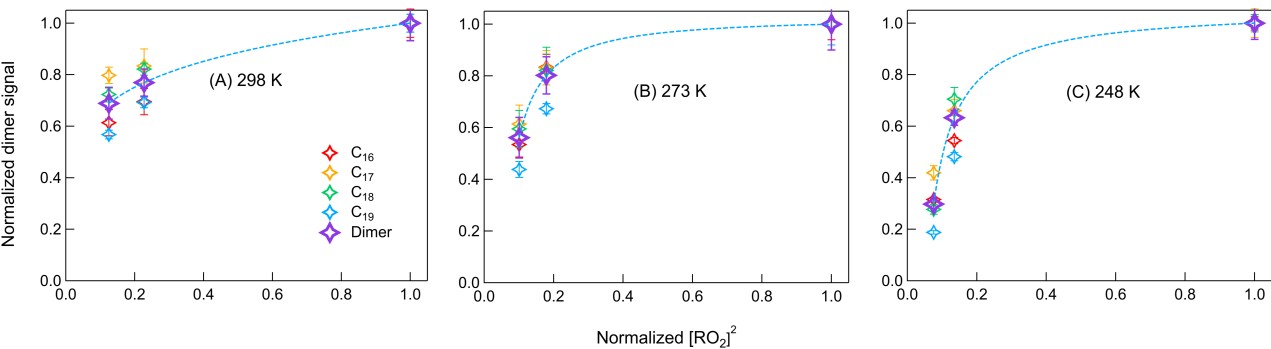

**Figure 9. Correlation between normalized dimer signals in both phases and normalized [RO$_2$]$^2$ at (A) 298 K (B) 273 K (C) 248 K (Exp. 298abc, 273abc, 248abc). The dashed lines represent the changing tendency of dimers.**

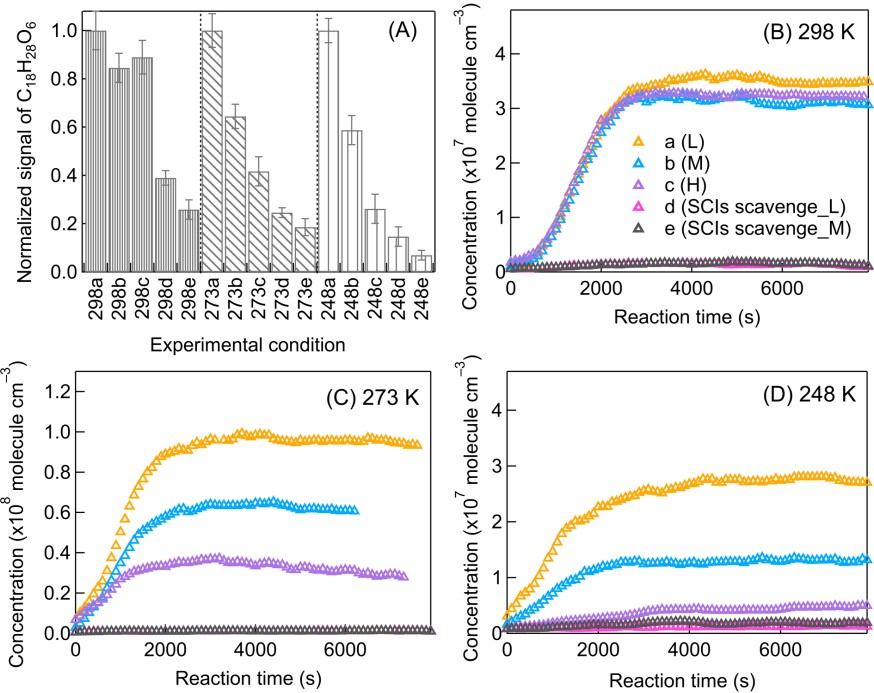

**Figure 10.** The impact of [HO$_2$]/[RO$_2$] and SCIs scavenging on (A) the total C$_{18}$H$_{28}$O$_6$ signal in both phases (normalized to the largest signal at each temperature, more than 90% from the particle phase for all conditions), and the gas-phase variation of C$_{18}$H$_{28}$O$_6$ at (B) 298 K (C) 273 K (D) 248 K (a−c: increasing [HO$_2$]/[RO$_2$]; d, e: scavenging SCIs).

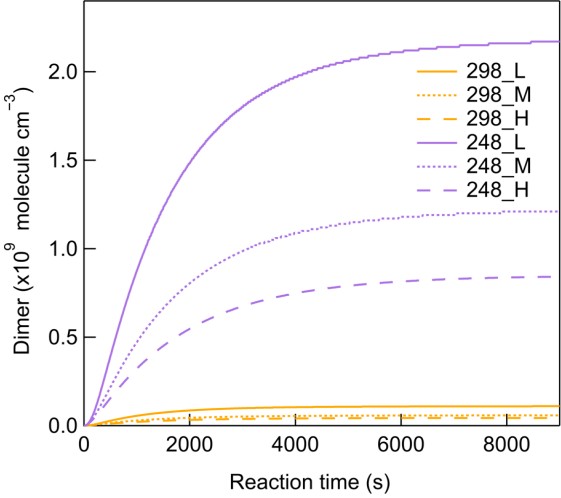

**Figure 11. Simulated dimers formation from reaction of C₉-SCIs with RO₂ radicals at different [HO₂]/[RO₂] conditions for 298 K and 248 K.**