# Peer review of "Impact of temperature on the role of Criegee intermediates and peroxy radicals in dimers formation from $\beta$ -pinene ozonolysis"

_EGUsphere, 2023_

## Author Comment (AC2)

**Response to reviewers' comments on "Impact of temperature on the role of Criegee intermediates and peroxy radicals in dimers formation from β-pinene ozonolysis"**

The authors greatly thank the reviewers for the careful review of our manuscript and the valuable feedback. All the comments are addressed point by point, with our responses in blue, and the corresponding revisions to the manuscript in red. All updates are marked in the revised manuscript. According to the comments, we have added new analysis to strengthen our work.

**Response to Reviewer #1**

Gong et al. studied the SOA formation of β-pinene ozonolysis at 298 K, 273 K and 248 K. Coupling the laboratory measurement and chemical modelling, the authors provide insights into the mechanisms of dimers formation in the β-pinene ozonolysis SOA at three studied temperatures. Considering the importance of SOA formation in winter or cold regions or high altitude, the topic is of high interest to the research community in atmospheric science. The study was well designed, but only a subset of data was analyzed and presented. The current discussion is mostly limited to the dimers in the particle phase. I would highly encourage the authors to carry out more comprehensive data analysis for the gas- and particle-phase data before they reach the conclusions. I would support the final publication after addressing my comments below.

**Major Comments**

1. SOA studies at low temperatures are increasingly popular (Huang et al., 2018; Ye et al., 2019; Simon et al., 2020; Gao et al., 2022; Gao et al., 2023), of which the findings inform the SOA formation at high altitudes or colder regions. In particular, several KIT studies (Huang et al., 2018; Gao et al., 2022; Gao et al., 2023) which used the same chamber as this study have reported the effect of temperature on the formation of biogenic SOA and their physicochemical properties. The authors should briefly summarize the findings of the previous studies at least from KIT and then discuss what new values this study is bringing.

We have summarized the previous findings on the temperature effect on SOA, and this part is included in the introduction section as follows.

Several studies reported higher SOA yields at lower temperatures in α-pinene oxidation (Jonsson et al., 2008; Pathak et al., 2007, 2008; Saathoff et al., 2009). In recent years, more attention has been paid to the temperature impact on aerosol constituents and physicochemical properties. Ye et al. (2019) and Simon et al. (2020) reported highly oxygenated molecules (HOMs) were less abundant at lower temperatures in α-pinene oxidation due to the positive temperature dependence of autoxidation reaction (Praske et al., 2017), however, the reduction of the saturation vapor pressure at lower temperatures counteracted the chemical effect on new particle formation. Kristensen et al. (2017) reported suppressed formation of dimers at subzero temperatures. Huang et al. (2018) studied the interactions between particle composition and viscosity at 223 K. Gao et al. (2021, 2023) investigated the temperature effect on the composition and volatility of aerosols from β-caryophyllene oxidation. These studies indicate that besides the impact on volatility and partitioning, temperature impacts the chemical reaction mechanism and product formation. Although the temperature impact on HOMs formation was studied, the understanding of the temperature impact on the formation pathways of other important SOA constituents such as dimers is limited. This study tries to bring some new insights into the reaction mechanisms of two kinds of reactive intermediates in the atmosphere: stabilized Criegee intermediates (SCIs) and organic peroxy radicals ($RO_2$) at lower temperatures, and their further impacts on dimers and SOA formation.

2. The $O_3$ concentration used in the experiments was approximately 1 ppm, which was higher than the typical ambient level. How are the findings of the study applicable to the real atmosphere? Please give a justification. To further strengthen the implications of the work, model simulations should be also carried out for a scenario of ambient level of $O_3$.

In order to consume most β-pinene in the experimental timescale, and to avoid the large wall loss due to the long residence time, the $O_3$ level used in this chamber study was much higher than the typical ambient level. We have clarified this in the experimental section as follows.

It should be noted that the $O_3$ level used in this chamber study was much higher than the typical ambient level for two reasons: first, for better comparison among different conditions more than 90% of β-pinene was expected to be consumed in each experiment; second, a long residence time was avoided to reduce the impact of wall losses.

We also added model simulations at the ambient $O_3$ level, and a discussion on transferring the findings from this chamber study to the real atmosphere was added in section 4.4.

To better transfer the findings in this study to the real atmosphere, box model simulations at ambient $O_3$ level of 40 ppb, and β-pinene of 2 ppb were conducted at 298 K and 248 K. The modeling for more atmospheric conditions focused on $O_3$ reaction, and the OH concentration was set at a low level ($1\times10^4$ molecule $cm^{-3}$). Different $HO_2$ concentrations of $2\times10^7$ molecule $cm^{-3}$, $1\times10^8$ molecule $cm^{-3}$, and $5\times10^8$ molecule $cm^{-3}$ were used for deriving different $[HO_2]/[RO_2]$ as 0.05–0.1, 0.8–2.5, and >20. Similar mechanisms elaborated in Section 3 were utilized and the proposed reaction coefficients of SCIs were utilized at 248 K. The simulations lasted for 7 days, leading to that the accumulated oxidized β-pinene was 23.5 ppb at 298 K and 12.7 ppb at 248 K. The results showed that with $[HO_2]/[RO_2]$ of 0.05–0.1, accumulated concentrations of dimers

formed from $C_9$-SCIs with $RO_2$ were $2.26 \times 10^7$ molecule $cm^{-3}$ at 298 K and $1.18 \times 10^8$ molecule $cm^{-3}$ at 248 K. With $[HO_2]/[RO_2]$ increasing to 0.8–2.5 and >20, dimers formed from this channel decreased to $6.57 \times 10^6$ molecule $cm^{-3}$ and $1.28 \times 10^6$ molecule $cm^{-3}$ at 298 K, and $2.44 \times 10^7$ molecule $cm^{-3}$ and $4.28 \times 10^6$ molecule $cm^{-3}$ at 248 K. The atmospheric relevant simulations demonstrated that the variation of temperature and $[HO_2]/[RO_2]$ can have a significant influence on the dimers formation also for ambient $O_3$ levels. It should be noted that in the current simulation, only β-pinene-derived $RO_2$ were considered, while in the real atmosphere, the $C_9$-SCIs have opportunities to react with $RO_2$ formed from other VOCs.

3. If the instrumentation was operated at a temperature which was higher than that in the AIDA chamber, potential losses of particle-phase compounds due to evaporation could occur in the tubing. Please justify how the temperature difference affected the observed gas and particle components in detail.

This is an issue that needs to be considered for low temperature experiments. We added a discussion on this issue in the experimental section.

Since all the instruments and the filter sampling were operated at 295±2 K, the influence of the temperature difference to the simulation chamber needs to be considered (Gao et al., 2023). For the online measurement instruments, a bypass flow was added to reduce the residence time in the sampling tubes to be less than 10 s. Such a short residence time avoids significant particle evaporation and diminishes artifacts on online measurements. For particles collected on PTFE filters, which were later analyzed by FIGAERO-CIMS, the sampling time of each filter was 15–20 minutes. Due to the short residence time of about 1 s in the sampling line the filter was significantly cooled for low temperature experiments. Before storing the filter samples in a freezer at 253 K there were also 5–10 minutes of handling time. Hence, we cannot rule out that some particulate compounds could evaporate during the sampling, resulting in a potential underestimation of some more volatile compounds in the particle phase for low temperature experiments. However, considering the dry conditions of our experiments, substantial evaporation of (semi-)volatile compounds from particle phase should be hindered due to the high viscosity of the particles.

4. Lines 220–231 and Lines 242–244: The authors are encouraged to provide more insights into how $[HO_2]/[RO_2]$ affect the SOA formation at different temperatures. Given the gas-phase CIMS measurement was available in the study, the authors should provide detailed analysis on how different $[HO_2]/[RO_2]$ affects the gas phase composition, volatility distributions and consequently SOA formation.

The gas-phase composition from CIMS measurement for different $[HO_2]/[RO_2]$ and temperatures was provided in the Supplement (Fig. S9). We also added a detailed analysis on the effect of $[HO_2]/[RO_2]$ on gas-phase composition and volatility distribution in section 4.1.

$[HO_2]/[RO_2]$ was changed by using different concentrations of CO, resulting in different concentrations of $HO_2$ radicals. Higher $HO_2$ concentrations led to a larger sink for $RO_2$ radicals and consequently a lower $RO_2$ concentration. Additionally, changing $[HO_2]/[RO_2]$ impacts the branching ratios of product formation from $RO_2$ reactions. Reaction with $HO_2$ is a chain termination for $RO_2$ radicals, leading to the formation of more volatile products, while reactions with $RO_2$ can be chain termination or chain propagation processes. Docherty and Ziemann (2003) proposed that the formation of pinic acid in β-pinene oxidation was inhibited by the increasing $[HO_2]/[RO_2]$ as the formation pathway of pinic acid involved a series of $RO_2$ chain propagation reactions. The inhibition of increasing $[HO_2]/[RO_2]$ on pinic acid formation was also observed in this study for all temperatures. The gas-phase composition shown in Fig. S9 demonstrates that most gas-phase products observed in this study are monomers and smaller molecules, while gas-phase dimers only account for a small fraction. Figure 4 shows the volatility distribution of gas-phase products at different $[HO_2]/[RO_2]$. Although increasing $[HO_2]/[RO_2]$ promoted the formation of several compounds, the suppression of increasing $[HO_2]/[RO_2]$ on gas-phase products observed from CIMS was more obvious. The relative inhibition of increasing $[HO_2]/[RO_2]$ on gas-phase products was more significant at 273 K and 248 K compared to 298 K. The shift of the volatility classes with decreasing temperature results in the shifts of some gas-phase compounds to lower volatility classes. This could partly explain the larger inhibition effect of increasing $[HO_2]/[RO_2]$ on SOA yield at lower temperatures.

[Figure]

**Figure S9. The averaged CIMS gas-phase mass spectra for all temperatures at different [HO₂]/[RO₂] (Signals are normalized to total gas $C_xH_yO_z$, Exp. 298abc, 273abc, 248abc).**

[Figure]

**Figure 4. Volatility distribution of gas-phase products for different temperatures and [HO₂]/[RO₂] (Exp. 298abc, 273abc, 248abc). Markers are sized by the square root of their relative abundance and colored by the ratio of signals at different [HO₂]/[RO₂] versus signals at low [HO₂]/[RO₂] at each temperature. From left to right, colored bands in the background represent the volatility classes of ULVOC, ELVOC, LVOC, SVOC, and IVOC. These volatility classes are defined for 298 K and shift with temperature according to the Clausius-Clapeyron equation.**

5. Line 235: The monomers contribute to 54%–64% of the particle-phase signals and thus were important for the SOA formation in this study. However, there is little discussion about the monomers here. The authors need to provide more details about monomers in the context of SOA composition.

Indeed, also the monomers were important for SOA formation and therefore we added the following sentences to section 4.2.

As for monomers, the $C_8$ and $C_9$ products were major contributors, which accounted for about 55% and 35% of monomers. $C_{10}$ products usually accounted for less than 10% of total particulate monomers because $C_{10}$ monomers mainly formed from β-pinene reaction with OH. The fractions of $C_{8-10}$ monomers in total particulate composition were similar under different temperature conditions.

6. Lines 243–244: Please provide more discussions about how their gas-phase formation pathways were influenced with the use of the gas-phase data from CIMS.

The discussion about the influence of increasing $[HO_2]/[RO_2]$ and SCIs scavenging on gas-phase formation pathways of dimers was added to section 4.2.

The impact of $[HO_2]/[RO_2]$ on gas-phase dimers is shown in Fig. S11, demonstrating the dimers formed from $RO_2+RO_2$ reactions in the gas phase were inhibited with increasing $[HO_2]/[RO_2]$. The relative inhibition was more significant at 273 K and 248 K than at 298 K. Scavenging SCIs resulted in significant suppression on gas-phase dimers for all temperatures as shown in Fig. S12, indicating that dimers formation from gas-phase SCIs reactions were hindered. Since the volatilities of these dimers are presumably sufficiently low enough, they should be primarily in the particle phase even at 298 K, which is in agreement with the FIGAREO-CIMS measurement in both phases. The changing of gas-phase formation pathways of dimers also had a significant impact on particulate dimers, which was shown in Fig. S13.

[Figure]

**Figure S11. Mass defect plots of gas-phase dimers at different temperatures. Markers are sized by the square root of their signals and colored by the ratio of signals at middle and high $[HO_2]/[RO_2]$ versus signals at low $[HO_2]/[RO_2]$ at each temperature (Exp. 298bc, 273bc, 248bc).**

[Figure]

**Figure S12. Mass defect plots of gas-phase dimers at different temperatures at low [HO$_2$]/[RO$_2$]. Markers are sized by the square root of their signals and colored by the ratio of signals with or without FA versus signals without FA at each temperature (Exp. 298ad, 273ad, 248ad).**

7.  Lines 246–249: The authors claim that there is a linear relationship between the particle-phase dimers and monomers in the particle phase, consistent with the findings of Zhao et al. (2018). The analysis of this study was carried out against the particle phase, which is totally different from the one done by Zhao et al. (2018) (i.e., the gas phase). It will be more appropriate if the gas-phase analysis is provided here. In addition, Pospisilova et al. (2020) have observed rapid condensed-phase reactions in the α-pinene ozonolysis SOA under 40%–50% RH conditions at 295 K. Different from Pospisilova et al. (2020), the experiments presented in this study were performed at low RH and/or low temperature. Under such conditions, particle viscosity can be substantially high and potentially affect the diffusion of organic molecules (e.g., monomers) in the particle phase. Please provide a discussion about the impact of particle viscosity on the particle phase reactions forming dimers.

We are aware of the difference of the analysis between this study and Zhao et al. (2018). The correlation between monomers and dimers in the gas phase is shown in Fig. S14. We added a discussion on the impact of particle viscosity on the dimer's formation in the particle phase as follows in the section 4.2.

The linear correlation between monomers and dimers in the gas phase shown in Fig. S14 indicates that the dimers identified here are not formed from the clustering of closed-shell monomers. Similar observations were reported for α-pinene oxidation (Zhao et al., 2018). Some particle-phase reactions are reported to influence the formation and decomposition of dimers (Pospisilova et al., 2020; Renbaum-Wolff et al., 2013). Most of our experiments were conducted at dry conditions, leading to a relatively high viscosity of the aerosols and slow particle-phase diffusion. Hence, the potential impact of particle-phase reactions on dimers was limited. We also observed a linear correlation between the particle-phase dimers and monomers as shown in Fig. S15, which suggested that the contribution of particle-phase clustering of monomers to the dimers observed in this study was limited. Although the contribution of particle-phase reactions on dimers cannot be excluded completely, our observations can be explained well by gas-phase reactions.

[Figure]

**Figure S14. The correlation between gas-phase dimers and monomers at different temperatures (Exp. 298a, 273a, 248a). Solid lines are linear fit with R$^2$ larger than 0.9.**

8.  Lines 256–257: The volatilities of C17–19 dimers are presumably sufficiently low enough and thus they should be primarily in the particle phase even at 298 K. Please quantitatively estimate how much the increase in C17–C19 dimers in particles can be attributed to the decease of volatilities but also the reduced wall losses when temperature decreased from 298 K to 273 K.

Considering that most of dimer species observed in this study stay in the particle phase, the wall loss effect and the influence of changing volatility with decreasing temperature on these dimers are different from those smaller molecules. We added a discussion on this issue in the section 4.2.

The gas-phase dimers account for less than 5 % of the total gas- and particle-phase dimer signals, which means that most dimers have sufficiently low vapor pressures and primarily stay in the particle phase even at 298 K. Due to their low gas phase concentrations potential wall losses via the gas phase can have only a small impact. Besides, the wall loss of particle mass was calculated to be usually below 10%, suggesting the wall loss effect on the particulate dimers can be regarded as of limited importance. Therefore, changes of these particulate dimers with temperature can be mainly attributed to the impact of temperature on their formation pathways.

9.  Lines 270–281: Instead of analyzing selective compounds, the authors should carry out more thorough analysis with the thermograms. As discussed, the temperature and/or [HO$_2$]/[RO$_2$] conditions can affect the production of SVOCs and LVOCs in the manuscript. If so, the particle volatilities changed accordingly. In addition, it is possible that for

an identified ion, its desorption temperature varied with the experimental conditions, if isomers of different volatilities were produced. Therefore, it will be useful to add a) the total ion thermograms of particle samples; and b) statistics about the desorption temperatures of all observed ions.

The total ion thermograms are shown in Fig. S17. We also summarized the $T_{max}$ values of all the monomers and dimers at different temperatures in the Supplement (Table S2). It is possible to derive more volatility information by using methods such as positive matrix factorization analysis (Buchholz et al., 2020). However, this is beyond the scope of this manuscript.

The sum thermograms at different temperatures are shown in Fig. S17, and the $T_{max}$ of all the monomers and dimers are summarized in Table S2.

[Figure]

**Figure S17. Sum thermograms at different temperatures with different [HO₂]/[RO₂] and SCIs conditions (Exp. 298acd, 273acd, 248acd).**

**Minor Comments**

1. The abstract is too long and descriptive for the readers. It needs to be more concise and compact for better readability.

We have revised the abstract to make it more concise.

**Abstract.** Stabilized Criegee intermediates (SCIs) and organic peroxy radicals (RO₂) are critical in atmospheric oxidation processes and secondary organic aerosol (SOA) formation. However, the influence of temperature on their corresponding reaction mechanisms in SOA formation is unclear. Through utilizing formic acid as SCIs scavenger and regulating the ratio of hydroperoxyl radials (HO₂) to RO₂ ([HO₂]/[RO₂]) from ~0.3 to ~1.9 using different concentrations of CO, the roles of RO₂ and SCIs in SOA formation were investigated from 248 K to 298 K, particularly for dimers formation in β-pinene ozonolysis. The SOA yield increased by 21% from 298 K to 273 K, while decreased by 40% from 273 K to 248 K. Both changing [HO₂]/[RO₂] and scavenging SCIs significantly affect SOA yield and composition. SCIs reactions accounted for more than 40% of dimers and SOA mass formation for all temperatures. Increasing [HO₂]/[RO₂] inhibited dimers and SOA formation, and this inhibition became larger with decreasing temperature. Compared to low [HO₂]/[RO₂] (0.30–0.34), the dimers abundance at high [HO₂]/[RO₂] (1.53–1.88) decreased by about 31% at 298 K and 70% at 248 K. [HO₂]/[RO₂] has a specific impact on SCIs-controlled dimers at lower temperatures by influencing especially the C₉-SCIs reactions with RO₂. The dimers formed from C₉-SCIs reactions with RO₂ were estimated to decrease by 61% at high [HO₂]/[RO₂] compared to low [HO₂]/[RO₂] at 248 K. The high reactivity and substantial contribution to SOA of β-pinene-derived SCIs at lower temperatures observed in this study suggest that monoterpene-derived SCIs reactions should be accounted for in describing colder regions of the atmosphere.

2. Line 11: Specify what are the SCIs scavengers used in this study.

We have clarified the SCIs scavengers as follows.

Through utilizing formic acid as SCIs scavenger and regulating the ratio of hydroperoxyl radials (HO₂) to RO₂ ([HO₂]/[RO₂]) from ~0.3 to ~1.9 using different concentrations of CO...

3. Lines 14–15 and Lines 256–257: Whether the changes in SOA yield were partially attributed to partitioning as well as wall losses or not? Please give an estimation about how the changes in SOA yield was affected by partitioning and wall losses.

We have revised the paragraph in section 4.1 as follows.

The SOA yields are calculated as ratio of the SOA mass concentration ($\mu g \cdot m^{-3}$) versus the mass concentration of β-pinene reacted ($\mu g \cdot m^{-3}$). The SOA yields measured for different experimental conditions are shown in Fig. 3A. The aerosol dynamic model calculations of the particle lost and gases lost to the walls are shown in Fig. 3 for different temperatures. The results demonstrated that the particles lost to the walls accounted for less than 10% of the SOA mass concentrations at all temperatures within the timescale of the experiment. The mass of gases lost to the walls was largest at 298 K of about 25.5% in total SOA mass, and the wall loss effect of gases became smaller with decreasing temperature. The calculated SOA yield at 298 K and

low [HO$_2$]/[RO$_2$] was (12.8±1.0) %, and around 16.8% after wall loss correction, which was in the range of previously reported SOA yield for β-pinene ozonolysis (Table S1). When the temperature decreased to 273 K, the SOA yield increased to (15.5±1.1) % (19.0% after wall loss correction) at low [HO$_2$]/[RO$_2$]. With the temperature further decreasing to 248 K, the SOA yield decreased to (9.1±0.6) % (11.6% after wall loss correction).

4. Line 20: It will be good to mention that the [RO$_2$]$^2$ was derived from the model simulations.

We clarified this in the revised manuscript.

5. Line 37: Please briefly summarize the current knowledge about how temperature impacts the chemical reaction mechanisms regarding SOA constituents.

The temperature impacts on SOA constituents in previous studies were summarized and added in the introduction section. Please see the answer to major comment 1.

6. Lines 33–35: References are too old. There are a series of temperature studies (Huang et al., 2018; Ye et al., 2019; Simon et al., 2020; Gao et al., 2022; Gao et al., 2023) in the past few years. Please include up-to-date citations here.

We have included these references in the introduction section. Please see the answer to major comment 1.

7. Line 44: Please specify "several trace species".

Now it is specified in the revised version as follows.

SCIs, formed from alkene ozonolysis, perform as an efficient oxidant for several trace species, e.g., SO$_2$, NO$_x$, carboxylic acids, carbonyl compounds, etc., contributing to the formation of inorganic and organic aerosol components.

8. Line 57: It is unclear that whether "RO$_2$+RO$_2$ reactions" mean self or cross reactions here but also many places in the manuscript.

In this manuscript, RO$_2$+RO$_2$ reactions include the self- and cross-reactions of RO$_2$ radicals. We have clarified this where "RO$_2$+RO$_2$ reactions" was used for the first time.

As for RO$_2$+RO$_2$ reactions, which include the self- and cross-reactions of RO$_2$ radicals, the rate coefficients vary over a wide range…

9. Line 105: Why only semi-volatile products? How about compounds of low volatility?

We have revised it as "semi- and low-volatile products".

10. Section 2.2: How was the FIGAERO-CIMS calibrated? Please provide additional details.

More details on the calibration of FIGAERO-CIMS were provided in the Supplement as follows.

Pinic acid, as the most abundant product formed during the reaction, was utilized to characterize the sensitivity of CIMS. Pinic acid (C$_9$H$_{14}$O$_4$, 92%, Chemspace) was dissolved into methanol to $1.77\times10^{-4}$, $1.77\times10^{-5}$, and $1.77\times10^{-6}$ g mL$^{-1}$ as standard solutions. Different volumes of the standard solutions were deposited on a PTFE filter using a syringe. The filter was heated by FIGAREO-CIMS in ultra-high-purity nitrogen following a thermal desorption procedure. The results are shown in Fig. S2.

[Figure]

**Figure S2. Calibration of the FIGAERO-CIMS with pinic acid.**

11. Lines 129–130: What were the mass loadings of the collected filter samples? Delaying desorption on the FIGAERO which arises from too high mass loading can affect the thermogram analysis and interpretation (Huang et al., 2018).

This issue was clarified in the experimental section as follows.

The sampling time of each filter was typically 15–20 minutes adjusted to organic aerosol mass concentrations in the chamber to achieve sufficiently low and similar mass loadings on each filter of about 1 μg.

12. Lines 162–167: Model sensitivity tests should be carried out to investigate the impacts of negative or positive temperature dependence of $RO_2+RO_2$ reactions.

Model sensitivity tests for $RO_2+RO_2$ reactions were carried out and the impacts of different temperature dependences of $RO_2+RO_2$ reactions were clarified in section 3.1.

Here the model sensitivity tests for $RO_2+RO_2$ reactions in β-pinene oxidation were carried out. For the test of the negative temperature dependence of $RO_2+RO_2$ reactions, the reaction coefficients of $RO_2+RO_2$ reactions at 273 K and 248 K were modified as 1.5 and 2 times of those at 298 K referring to the variation of $HO_2+RO_2$ reaction coefficient at different temperatures. The results showed that compared to conditions without temperature impact on $RO_2+RO_2$ reactions, the negative temperature dependence led to a decrease of simulated $RO_2$ concentration for about 4% and 10% at 273 K and 248 K. Conversely, when the reaction coefficients of $RO_2+RO_2$ reactions at 273 K and 248 K were modified as 0.75 and 0.5 times of those at 298 K, the positive temperature dependence of $RO_2+RO_2$ caused an increase for 4% and 10% of simulated $RO_2$ concentrations at 273 K and 248 K. The modeling results showed that the changing $RO_2+RO_2$ reaction coefficients within 2 times for different temperature dependences could cause a limited effect on $RO_2$ concentrations.

13. Section 3.2: How were the volatilities and gas-particle partitioning were calculated in the model? What were the particle wall loss rates?

Volatilities were derived from the FIGAERO-CIMS measurements. The gas-particle partitioning was calculated based on the volatilities in the model. The particle wall loss rates were calculated based on the size-dependent sedimentation and diffusion.

14. Line 185: "indicating that the wall loss rates of organic acids were higher at higher temperature" is redundant.

We have deleted the sentence.

15. Line 201: Please comment why new particle formation still occurred.

We explained this in section 4.1.

Although seed particles were added as condensational sink, new particle formation still occurred due to the formation of low-volatile products with a strong nucleation capability.

16. Lines 202–203: Considering the appearance of new particle formation and potential size-dependent variations in the organic-to-inorganic ratios, it can be technically challenging to calculate the SOA density. I would suggest the authors provide detail descriptions to explain how they derived the SOA density.

More details on SOA density calculation were provided in the Supplement as follows.

Effective density of aerosol was derived from comparing the mass distribution versus vacuum aerodynamic diameter from AMS, and the size distribution versus mobility diameter from SMPS. Considering that these newly formed particles during reaction were usually smaller than 70 nm, which could not be measured by AMS, the effective density calculated here relied on the organics that partitioned to the inorganic seed particles.

17. Line 206: Was the uncorrected SOA mass concentration used to calculate the SOA yield?

We have clarified this as follows.

The calculated SOA yield at 298 K and low $[HO_2]/[RO_2]$ was (12.8±1.0) %, and around 16.8% after wall loss correction, which was in the range of previously reported SOA yield for β-pinene ozonolysis (Table S1).

18. Line 208: Were the previously reported SOA yield for β-pinene ozonolysis obtained at 298 K and low $[HO_2]/[RO_2]$? What is the range of the reported SOA yield?

A summary of the previously reported SOA yield from β-pinene ozonolysis and the corresponding reaction conditions was added in the Supplement.

**Table S1. SOA yields and reaction conditions in β-pinene ozonolysis studies.**

| Temperature (K) | RH (%) | OH scavenger | Aerosol mass (μg·m$^{-3}$) | SOA yield (%) | Reference |
|---|---|---|---|---|---|
| 263–303 | <0.03 | none | 2–284 | 3–39 | von Hessberg et al., 2009 |
| 300 | 22 | none | 156 ± 3 | 18 ± 1 | Xu et al., 2021 |
| 293 | 6.3 | cyclohexane | 174 ± 3 | 17 ± 1 | Lee et al., 2006 |
| 306–307 | – | 2-butanol | 11–19 | 4–8 | Yu et al., 1999 |
| 248–298 | <0.01 | CO | 18–25 | 11–19 | This work |

19. Line 210: Please estimate the extent to which the partitioning process was promoted due to the decrease in temperature. This can be simply done by comparing the model simulations with and without accounting for the temperate dependence of volatilities.

In our aerosol dynamic model, the gas-particle partitioning was calculated based on the volatility distribution from FIGAERO-CIMS. For different temperatures, the yields of products with different volatilities are different, which makes the significance of simulations with and without accounting for volatility dependence be limited. Please see the answer to minor comment 3.

20. Lines 216–219: Whether the temperature dependence of SOA formation was observed in SOA particles other than β-pinene ozonolysis ones in previous studies?

The temperature impact on SOA formation from α-pinene in previous studies was summarized in the introduction section. Please see the answer to major comment 1.

21. Line 219: "which contribute to the formation of SVOCs and LVOCs" is unclear. How does it link to the observed temperature dependence of SOA formation / yield?

We have removed the sentence.

22. Line 227: Please specify the unwanted interferences if high concentrations of FA were used.

We clarified this in the manuscript.

To avoid unwanted particle-phase reactions and interference on the gas-phase CIMS measurement caused by the extremely high FA concentration,...

23. Line 229: How was the scavenging efficiency of more than 70% estimated?

We added explanations on calculating SCIs scavenging proportion in the manuscript.

Since the reaction coefficients of β-pinene-derived $C_9$-SCIs are not clear, it is difficult to calculate the proportion of the scavenged SCIs directly. In SCIs scavenging experiments more than 70% of the gas-phase dimers were diminished for all temperatures, based on which we estimated that more than 70% of $C_9$-SCIs were scavenged.

24. Line 234: What are the other compounds in Fig S6? Depending on the experimental condition, they account for considerable fractions of the particle samples.

The other species in this figure include $C_{1-7}$, $C_{11-15}$, and $C_{21-29}$ compounds with formulas of $C_{1-7}H_{2-14}O_{2-9}$, $C_{11-15}H_{12-28}O_{3-11}$, and $C_{21-29}H_{30-56}O_{5-12}$. This is clarified in the caption now.

25. Line 235: What is the use of the mass defect plots here?

We have removed the plot.

26. Lines 236–237: I disagree that the formation mechanisms of C20 dimers were not discussed due to their lower abundance. The authors should briefly discuss why the production of C20 dimers was low across the experimental temperatures.

We have added explanations for the low abundance of $C_{20}$ dimers observed in this study.

The $C_{20}$ dimers had a lower abundance accounting for only ~ 5% of all dimers. This can be explained by the fact that the major SCIs and $RO_2$ formed from β-pinene ozonolysis contain 8 or 9 carbon atoms.

27. Line 270: What is the fraction of monomers showing two peaks at different temperatures?

We did not observe any obvious temperature impact on the fraction of monomers showing two peaks.

28. Line 284: It is unclear that how the authors came to the point that the contribution of β-pinene-derived SCIs to dimers formation through bimolecular reactions was significant.

We have modified the sentence as follows.

The scavenging of SCIs leads to a reduction of more than 40% in total dimers from 248 K to 298 K (Fig. 6), indicating the significant contribution of β-pinene-derived SCIs to dimers formation.

29. Lines 293–294: Provide a table in SI to show the molar yields of HCHO and nopinone and experimental temperature for this study and literatures.

We have summarized the molar yields of HCHO and nopinone in previous studies in the Supplement.

**Table S3. The molar yields of HCHO and nopinone in β-pinene ozonolysis studies.**

| Temperature (K) | OH scavenger | RH (%) | HCHO | Nopinone | Reference |
|---|---|---|---|---|---|
| 296±2 | cyclohexane | dry | 0.65±0.04 | 0.16±0.04 | Winterhalter et al., 2000 |
| 293 | cyclohexane | 6.3 | 0.65±0.06 | 0.17±0.02 | Lee et al., 2006 |
| 295±4 | cyclohexane | dry | – | 0.16±0.03 | Ma and Marston, 2008 |
| 306–307 | 2-butanol | – | – | 0.16–0.17 | Yu et al., 1999 |
| 248–298 | CO | <0.01 | 0.63±0.06 | 0.16±0.02 | This study |

30. Figure 6: Please indicate which experiments were included for the analysis here.

We have indicated the experiments used for this figure in the caption.

31. Line 329–334: Figure 6 shows that non-SCIs-controlled dimers are important at 298 K. It is confusing that how higher RH inhibited 40% of the dimer formation at 298 K, if the non-SCIs-controlled pathway is more important than the SCIs-controlled one at that temperature.

We have replotted this figure to avoid misunderstanding. For the inhibition of SCIs scavenger on dimers please see Figure 6.

[Figure]

**Figure 7. The relative changes of (A) SCIs-controlled and (B) non-SCIs-controlled abundant dimers at 298 K or 248 K versus 273 K (The relative standard deviations are within 25 %, Exp. 298a, 273a, 248a).**

32. Line 354: Please justify why "20%" and "40%" were chosen for the classification.

We removed the paragraph in the revised manuscript.

33. Figure S13: There is one additional marker beyond those listed in legend. What does the solid marker in purple represent?

We have removed the figure.

34. Lines 360–361: The variations in the temperature dependences are not obvious in Figure S13.

We have removed the figure.

35. Lines 362–363: Which figure shows the influence of the variation of [HO$_2$]/[RO$_2$] on C$_{18}$ dimers?

We clarified it in the revised manuscript.

It is intriguing to find that at low temperatures, the variation of [HO$_2$]/[RO$_2$] has such a big influence on C$_{18}$ dimers (Fig. 8)…

36. Line 366: The fractions from the gas and particle phase should be showed in Figure 9A.

The fraction of gas-phase intensity was less than 10% for all conditions. To make the figure more readable we clarified this in the caption. The revised figure is shown below.

[Figure]

**Figure 10. The impact of [HO₂]/[RO₂] and SCIs scavenging on (A) the total C₁₈H₂₈O₆ signal in both phases (normalized to the largest signal at each temperature, more than 90% from the particle phase for all conditions), and the gas-phase variation of C₁₈H₂₈O₆ at (B) 298 K (C) 273 K (D) 248 K (a−c: increasing [HO₂]/[RO₂]; d, e: scavenging SCIs).**

37. Lines 369–371: Please provide the time series of nopinone in the supporting information.

We provided the time series of nopinone in Fig. S20. This was clarified in the manuscript as follows.

The formation of nopinone, as the main product from SCIs reaction with CO, is not notably influenced by changing temperatures (Fig. S20), confirming that the CO consumption on C₉-SCIs is negligible.

[Figure]

**Figure S20. Time series of nopinone concentration at different temperatures (Exp. 298a, 273a, 248a).**

38. Line 374: It is unclear what are these two possible reasons here.

We clarified it in the revised manuscript:

After excluding the possible influence of CO and C₉H₁₄O₄…

39. Lines 387–388: Why 75 s⁻¹ and 15 s⁻¹ were chosen in the box model?

We clarified it in the section 4.4 as follows.

Although some studies reported the unimolecular reaction coefficients of simple SCIs, less is known about the unimolecular reactions of monoterpene-derived SCIs. Gong et al. (2021) estimated the unimolecular reaction coefficient of limonene-derived SCIs as 30 s⁻¹ and 100 s⁻¹ for different SCIs isomers at 298 K. According to this, we assumed that the unimolecular reaction coefficient of β-pinene-derived C₉-SCIs was 75 s⁻¹ at 298 K in modeling. The rate coefficients of SCIs unimolecular reactions are strongly influenced by temperature. It was reported that the unimolecular reaction coefficient of (CH₃)₂COO-SCIs increased by a factor of four with temperature increasing by 40 K (Smith et al., 2016). As for CH₃CHOO-SCIs, the unimolecular reaction coefficient increased by a factor of five with temperature increasing by 35 K (Robinson et al., 2022). Berndt et al. (2014) reported that the ratio of the (CH₃)₂COO-SCIs unimolecular reaction coefficient versus the reaction

coefficient with $SO_2$ increased by a factor of 34 from 278 K to 343 K. Based on these temperature dependencies, the unimolecular reaction coefficient of $C_9$-SCIs was assumed to decrease by a factor of 5 from 298 K to 248 K.

40. Line 407: There is little discussion about how temperature affect the compounds' volatilities in the manuscript.

Please see the answer to major comment 4.

41. Lines 409: Please specify what the positive- and negative-temperature-dependent processes are.

We revised the sentence as follows.

The SOA yield is not monotonic with decreasing temperature in β-pinene ozonolysis due to the joint influence of volatilities and chemical mechanisms.

42. Line 410: Provide citations for "Such influence with varying temperatures could also exist in other VOCs oxidation systems…".

We have modified the sentence as follows.

Such influence with varying temperatures may exist in other VOCs oxidation systems with higher SCIs yields contributing to SOA mass formation.

43. Line 413: Please clarify "… a constant yield is not sufficient to represent the SOA formation potential of β-pinene in models".

We have modified the sentence as follows.

The SOA formation potential of β-pinene is influenced by several parameters in the atmosphere, such as temperature, RH, and $[HO_2]/[RO_2]$, which are to a large extent correlated to the chemistry of CIs and peroxy radicals. Therefore, these parameters need to be accounted for to represent the SOA formation potential of β-pinene in atmospheric models.

**Technical Comments**

1. It will be better to use "L", "M", and "H" to indicate "low", "middle" and "high" in Figures. E.g., 298L, 298M, 298H.

We have modified the figures according to the suggestion.

2. Lines 18–19: "Increasing $[HO_2]/[RO_2]$ inhibited dimers and SOA formation with a higher sensitivity at lower temperatures." What is the meaning of higher sensitivity in this sentence?

We have rewritten the sentence as follows.

Increasing $[HO_2]/[RO_2]$ inhibited dimers and SOA formation, and this inhibition became larger with decreasing temperature.

3. Line 37: It should be "volatilities".

Yes, we have revised it.

4. Line 40: Please use another word instead of "performances".

We changed it as "impacts" in the revised version.

5. Line 43: "fate" should be plural here.

Yes, we have revised it.

6. Line 66: It should be "low volatilities".

Yes, we have revised it.

7. Lines 68 – 70: The sentence is too long to follow.

We have rewritten the sentence as follows.

Some particle-phase reactions, including hemiacetal reactions of peroxides and carbonyls, noncovalent clustering of carboxylic acids, and aldol condensation reactions, could contribute to dimers formation (Kenseth et al., 2018; Yasmeen et al., 2010). However, these pathways were not able to adequately explain the dimers observed, and gas-phase reaction pathways were proposed to be important (DeVault and Ziemann, 2021; Hasan et al., 2021).

8. Lines 79 – 81: Provide references.

We provided references in the revised manuscript:

The oxidation of α-pinene has been broadly investigated, however, different isomers of monoterpenes have different reaction mechanisms due to their molecule structures (Jenkin, 2004; Lee et al., 2006).

9.  Line 110: Please replace "low-oxidized products" with "lightly oxidized products".

Yes, we changed it as "lightly oxidized products".

10. Lines 175 – 177: Provide the wall loss rate of β-pinene although negligible.

We added more explanations for the wall loss of β-pinene.

For β-pinene, we observed the time variation of β-pinene before adding $O_3$, and the concentration of β-pinene remained constant for two hours.

11. Line 188: It should be "the impact of vapor wall losses".

Yes, we have revised it.

12. Line 197: Please indicate that Figure 2 only shows the timeseries data for 298 K in the text. In addition, same types of plots should be made for other experimental conditions but shown in the supporting information.

We have clarified this in the manuscript and the same type of plots for 273 K and 248 K are now added in the Supplement.

[Figure]

**Figure S7.** Time series of (A) β-pinene mixing ratio, $O_3$ mixing ratio, measured SOA mass concentrations, and simulated SOA mass concentrations after wall loss correction (B) Particle mass size distributions (dM/dlog$D_p$) at 273 K (Exp. 273a).

[Figure]

**Figure S8.** Time series of (A) β-pinene mixing ratio, $O_3$ mixing ratio, measured SOA mass concentrations, and simulated SOA mass concentrations after wall loss correction (B) Particle mass size distributions (dM/dlog$D_p$) at 248 K (Exp. 248a).

13. Line 206: The abbreviation $Y_{SOA}$ is redundant and only used for one time.

We have removed the abbreviation.

14. Line 210: It should be "gas-to-particle partitioning".

Yes, we have revised it.

15. Figures 5 and 7: Please increase the gaps between two neighboring categories.

We have modified these two figures.

16. Line 283: It should be "reduction" instead of "reduce".

We have revised it.

17. Line 306: Specify "which was critical for the results".

Sorry for the mistake. We mean this was not critical for the results. We have corrected this in the manuscript.

18. Lines 291–293: Please add plots showing the formation of HCHO and nopinone as a function of the consumed β-pinene for the 248 and 273 K conditions in Figure S11.

We added the plots of HCHO and nopinone formation as a function of the consumed β-pinene at 273 K and 248 K in Fig. S18. It was clarified in the manuscript as follows.

Figure S18 shows the formation of HCHO and nopinone as a function of reacted β-pinene at different [HO₂]/[RO₂] for all temperatures. Both the formation of HCHO and nopinone show good linear correlation with β-pinene reacted. Different [HO₂]/[RO₂] caused by different CO concentrations did not influence HCHO and nopinone formation, confirming that the CO reaction with SCIs was negligible in this study. No obvious temperature impact on HCHO and nopinone formation was observed, indicating that temperature did not influence the early reaction steps of CIs' generation.

[Figure]

**Figure S18. The formation of formaldehyde (HCHO) and nopinone as a function of β-pinene reacted at 298 K, 273 K, and 248 K for different [HO₂]/[RO₂] (Exp. 298abc, 273abc, 248abc). The lines represent linear fits and R² values are larger than 0.95.**

Lines 319 – 320: Please indicate that the dimers are the "chosen" or "selective" dimers.

We have described them as "selective dimers".

19. Line 333: What is the meaning of "less water effects"?

We have revised the sentence as follows:

…suggesting that the contribution of β-pinene to atmospheric SCIs and dimers could be more important in colder regions because of the lower water vapor concentration.

20. Figure 9: The color codes are confusing. The colors used for a (L), b(M), and c(H) in subplots (B-D) should be different from those used for different temperatures in the subplot (A).

We have modified the figure. Please see the answer to minor comment 36.

21. Lines 371 and 372: The sentence "The [HO₂]/[RO₂] impact… from C₉-SCIs reactions." is unclear. Please rephrase the sentence.

We have rewritten the sentence as follows.

C₉-SCIs contribute to the formation of C₁₈H₂₈O₆ mainly through reacting with C₉H₁₄O₄ in the gas phase, and the gas-phase concentrations of C₉H₁₄O₄ at different [HO₂]/[RO₂] needed to be compared.

22. Lines 377 – 379: The sentence "Since the interference of water vapor… and concentration of SCIs" is too long to be understood.

We have rewritten the sentence as follows.

During the reaction, the unimolecular reaction of SCIs, including isomerization and decomposition, was the main sink of SCIs and was crucial for determining the lifetime of SCIs.

23. Can you use "CIs" instead of Criegee intermediates throughout the manuscript after the abbreviation was introduced? Same applies for RO₂.

We used "CIs" instead of Criegee intermediates in the revised manuscript.

24. The full name of HO$_2$ need to be introduced when it is mentioned at the first time in the abstract as well as the main text.

The full name of HO$_2$, hydroperoxyl radial, is now mentioned at the first time in the abstract as well as the main text.

**References:**

Buchholz, A.; Ylisirniö, A.; Huang, W.; Mohr, C.; Canagaratna, M.; Worsnop, D. R.; Schobesberger, S.; Virtanen, A. Deconvolution of FIGAERO−CIMS thermal desorption profiles using positive matrix factorisation to identify chemical and physical processes during particle evaporation. Atmos. Chem. Phys. 2020, 20, 7693−7716.

[revised manuscript text omitted]

**Response to Reviewer #2**

This manuscript presents the results from a series of chamber experiments designed to investigate the formation of dimers from stabilized criegee intermediates (SCIs) as well as peroxy radicals (RO₂). The presented experimental protocol for the chamber experiments appears well-considered, and the presented experimental data is of great quality. This work also involves box-modelling of the chamber experiments to investigate aspects of the system not able to be directly measured. Again, the modelling approach employed here seems sound, with good model-measurement agreement where presented.

The topic of dimers' contribution to SOA is of increasing interest and the results presented in this manuscript will be useful to a range of researchers when considering the implications of such processes in ozone-impacted environments. The work is of high quality and many of the comments below simply address the presentation of the results. However, there are some potential weaknesses in the analysis, particularly when attempting to provide information on the formation pathways of dimers.

I support the publication of this manuscript after the following comments are addressed.

CIMS Calibrations:

Throughout the manuscript, the authors discuss the concentrations of various compounds measured with I-CIMS. The authors note in the experimental section that they calibrated the CIMS data to pinic acid. This is a reasonable approach given the lack of available standards for the vast majority of compounds discussed, but I believe that the authors should better highlight the large impact that this could have on the reported concentrations. Previous work has shown that compound responses to I-CIMS can vary by orders of magnitude, even for similarly structured compounds (Lee et al., 2014). The authors should note this at the end of Section 2.2. This calibration issue can also be problematic if the species distribution of a specified group changes as the result of a change in experimental conditions. For example, in Figure 4, each C16 compound will have a different I-CIMS response, so if the distribution of these compounds changes when the temperature changes (as the authors argue), then the average response of the C16 group will change. This could increase or decrease the observed drop in dimer signal depending on the change in average response of this group.

These calibration issues are difficult to overcome, and the approach the authors have taken is reasonable. However, the authors should discuss the potential effect of calibration issues where possible, or at least acknowledge the potential issue.

We would like to thank the reviewer for pointing out this issue. We already noted this issue in section 2.2, and more details about the CIMS calibration with pinic acid are provided in the Supplement.

Pinic acid ($C_9H_{14}O_4$), as the most abundant product formed during the reaction, was used to calibrate the CIMS, and a sensitivity of $12.6\pm1.5$ cps ppt$^{-1}$ for $10^6$ cps I$^-$ was observed. More details about the calibration can be found in the Supplement. In the following, we assumed the same sensitivity for all compounds detected by CIMS and used signal intensity for the comparison. It should be noted that the sensitivity of pinic acid may not represent the sensitivities of all compounds measured by CIMS. Compound responses to I$^-$-CIMS can vary by orders of magnitude, even for similarly structured compounds (Lee et al., 2014). This calibration issue can be problematic if the species distribution of a specified group changes as a result of the changing experimental conditions.

Box Models and Mechanisms:

Section 3 outlines the procedure used for modelling in this work, however there are several points in the manuscript that make it unclear how many models were run and what mechanisms were used. The authors should use Section 3 to consolidate a description of all of the model runs performed and any alterations to the MCM.

Line 295 – From here, the authors note a number of modifications to the MCM. It is not clear where these changes are used. Are all of these changes used in the models of each experiment (e.g., the data presented in Figure 1)? If so, then these changes should be noted in Section 3 when the mechanism is introduced. Otherwise, the changes should be described as "proposed changes", or something similar, to make it clear that this chemistry isn't implemented in the current models. Additionally, if these changes are not implemented in the current models, then the authors may consider implementing them to test the effect on model predictions (e.g., by reproducing Figure 11 with the added chemistry).

Line 376 – Here, there is mention of "a box model" but it isn't clear whether this is a separate set of models or the same models as before.

Lines 385 and 391 – New reactions and rates are listed here but it isn't clear which models they're used for. Again, if these are implemented in all of the model runs presented then they should be described alongside the original mechanism description in Section 3.

We consolidated the description of all model runs as suggested in section 3.1.

Considering the previously reported Criegee intermediates (CIs) formation in β-pinene ozonolysis and the measurement results from this study, some modifications were applied to the formation of β-pinene-derived CIs, which was elaborated in Section 4.3. By implementing these updates on CIs, the yield of stabilized CH₂OO decreases from 0.15 to 0.1, and the yield of C₉-

SCIs increases from 0.1 to 0.32. The OH yield from β-pinene ozonolysis decreases slightly from 0.35 to 0.3. It should be noted that for β-pinene SCIs, only reactions with CO, NO, NO$_2$, SO$_2$, and H$_2$O are concluded in MCM. When CO was used to adjust [HO$_2$]/[RO$_2$] by reacting with OH radicals, the possibility of CO reacting with SCIs needed to be estimated. The reaction coefficients of SCIs with CO are usually reported as smaller than 10$^{-18}$ cm$^3$ molecule$^{-1}$ s$^{-1}$ (Eskola et al., 2018; Kumar et al., 2014, 2020; Vereecken et al., 2015), and the reaction coefficient of 10$^{-18}$ cm$^3$ molecule$^{-1}$ s$^{-1}$ was applied for SCIs reaction with CO in the current model. For better simulating C$_9$-SCIs, the unimolecular reaction with a coefficient of 75 s$^{-1}$ and reactions with HO$_2$ and RO$_2$ with coefficients of 2×10$^{-11}$ cm$^3$ molecule$^{-1}$ s$^{-1}$ were implemented in the model, which was elaborated in Section 4.4. These modifications showed a limited impact on HO$_2$ and RO$_2$ concentrations. The modeling results shown below were derived by implementing these updates.

We revised the paragraph in the section 4.4 as follows.

The potential contribution of C$_9$-SCIs reaction with RO$_2$ radicals to dimers formation was evaluated by modeling. Simulations at 298 K were regarded as the basic scenario using the model described in Section 3. Simulations at 248 K were chosen for comparison by implementing the proposed reaction coefficients described below.

Use of Zhao et al. 2018:

In Section 4.2, the authors attempt to use the methodology of Zhao et al. 2018 to show that the measured particle-phase dimers do not result from the reaction of particle-phase monomers. There are three issues with the authors' use of Zhao et al. here:

Zhao et al. are discussing the formation of dimers in the gas phase from RO$_2$ radicals. This means that the application of this methodology to the particle phase with closed-shell products must be suitably justified.

A lack of a quadratic relation does not preclude the formation of the dimers from the monomers. I believe that such a quadratic relationship in Zhao et al. suggests the formation of dimers via RO$_2$ due to a second-order formation process represented by RO$_2$+RO$_2$-->ROOR and a first-order loss process such as rapid uptake to the particle phase. So, if the dimers were formed from the monomers via multiple reaction steps, or there were additional losses of the particle-phase dimers (as is likely to be true in the particle phase), then this quadratic relationship would not hold.

The presence of a linear relationship does not preclude the formation of the dimers from the monomers. Zhao et al. state that their observation of a linear relationship shows that "the dimers identified here are not due to a simple association (i.e., clustering) of closed-shell monomers." This cannot be generalized to a general statement on the formation of dimers in the particle phase without significant justification.

We agree and are aware of the difference of the analysis between this study and Zhao et al. (2018). The correlation between monomers and dimers in the gas phase is shown in Fig. S14. We added a discussion on the impact of particle viscosity on the dimer's formation in the particle phase as follows in the section 4.2.

The linear correlation between monomers and dimers in the gas phase shown in Fig. S14 indicates that the dimers identified here are not formed from the clustering of closed-shell monomers. Similar observations were reported for α-pinene oxidation (Zhao et al., 2018). Some particle-phase reactions are reported to influence the formation and decomposition of dimers (Pospisilova et al., 2020; Renbaum-Wolff et al., 2013). Most of our experiments were conducted at dry conditions, leading to a relatively high viscosity of the aerosols and slow particle-phase diffusion. Hence, the potential impact of particle-phase reactions on dimers was limited. We also observed a linear correlation between the particle-phase dimers and monomers as shown in Fig. S15, which suggested that the contribution of particle-phase clustering of monomers to the dimers observed in this study was limited. Although the contribution of particle-phase reactions on dimers cannot be excluded completely, our observations can be explained well by gas-phase reactions.

[Figure]

**Figure S14. The correlation between gas-phase dimers and monomers at different temperatures (Exp. 298a, 273a, 248a). Solid lines are linear fit with R$^2$ larger than 0.9.**

Correlation with $[RO_2]^2$:

In Section 4.4, the authors attempt to show that the gas-phase dimers measured are not formed predominantly via $RO_2$ by correlating the dimer signal with $[RO_2]^2$. A lack of linear correlation is taken as evidence that this is not the case. The authors should explain why they expect a linear correlation with $[RO_2]^2$, or reference this methodology in another paper, as it is not currently clear.

In fact, this analysis seems to be opposed to the analysis presented in Zhao et al. 2018 which has been previously misinterpreted. Zhao et al. plot $RO_2$ concentrations against dimer concentrations (in the same way as in Section 4.4) and use the quadratic relationship as evidence of ROOR formation from $RO_2$. Figure 8 does not appear to show a quadratic relationship as in Zhao et al., so the authors should attempt to explain this discrepancy.

We added more discussion on the correlations of dimers with $[RO_2]^2$ observed in section 4.4. The purpose of Fig. 9 was not going to prove the importance of $RO_2+RO_2$ for dimers formation as Zhao et al. (2018) already showed evidence for that. In this study, we would like to show different impacts of $[HO_2]/[RO_2]$ on dimers formation at different temperatures.

If $RO_2$ radicals influence dimers formation predominately through $RO_2+RO_2$ reactions, which are second-order reactions and should therefore lead to a linear correlation of dimer signal with $[RO_2]^2$. Zhao et al. (2018) showed a quadratic relationship between the gas-phase signals of dimers and $RO_2$ as evidence of dimers formation from $RO_2+RO_2$ reactions in α-pinene ozonolysis. In this study, we would like to show different impacts of $[HO_2]/[RO_2]$ on dimers formation at different temperatures. Figure 9 shows the correlations between dimers in both phases and simulated $[RO_2]^2$ at different temperatures. At 298 K, the dimer signals show a nearly linear relationship with $[RO_2]^2$. When $[RO_2]^2$ decreased by about 85%, dimers formation was inhibited for around 30%, indicating that the contribution of other pathways to dimers was also important in β-pinene ozonolysis at 298 K. The correlations between dimers and $[RO_2]^2$ at 273 K and 248 K became different from that at 298 K, suggesting that at lower temperatures there were different impacts of $RO_2$ on dimer formation pathways. Below we will discuss the possible reasons for the specific impact $[HO_2]/[RO_2]$ on dimers at lower temperatures.

**Additional Comments:**

1. Line 148– "…this reaction was calculated in the model and was regarded as too slow to make a difference to SCIs reactions". It is unclear how the authors determined this to be true. Were these rates implemented into the mechanism and shown to be insignificant? If so, then some data should be presented from such a model run to illustrate the insignificance. Alternatively, if this conclusion is simply reached by the magnitude of a rate calculated with typical CO and SCI concentrations being small then this should be stated along with the calculated rate.

We have clarified this in the manuscript.

…and the reaction coefficient of $10^{-18}$ cm$^3$ molecule$^{-1}$ s$^{-1}$ was applied for SCIs reaction with CO in the current model…The results showed that reaction with CO accounted for less than 1% of SCIs at all temperatures.

2. Line 176– "…and found that the wall loss of β-pinene was negligible at all temperatures". The authors should provide some indication of the magnitude of this loss (even if it is very small) either via a calculated loss rate or a plot of the concentrations not changing over a reasonable time period. The same is true for nopinone and HCHO discussed in the following sentence.

We added more explanations for the wall loss of β-pinene, nopinone, and HCHO.

For β-pinene, we observed the time variation of β-pinene before adding $O_3$, and the concentration of β-pinene remained constant for two hours.

Two abundant carbonyl products, nopinone and formaldehyde (HCHO), were measured and their concentrations remained constant for two hours.

3. Line 179– Calculating the $O_3$ loss rate after β-pinene was added to the chamber introduces a potential artifact resulting from the reaction of ozone with secondary oxidation products as opposed to just the chamber walls. Do the authors have data showing the wall loss of $O_3$ in the absence of any organic compounds (either before β-pinene was added to the chamber or in a separate experiment where no β-pinene was added to the chamber)? If not, then the authors should justify why it is reasonable to calculate the $O_3$ loss rate in the presence of organic compounds, such as the oxidation products of β-pinene.

We clarified this in the revised manuscript as follows.

Since there is one unsaturated bond in β-pinene molecule, the further $O_3$ oxidation on the products was regarded as negligible. This procedure was supported by experiments where similar ozone wall loss rates were measured in the absence of other compounds.

4. Line 182– How was the loss of FA and $C_9H_{14}O_4$ calculated? From Figure S5 it seems like the authors introduced the species into an empty chamber and then observed the decay. If so, this should be explained.

We clarified this in the revised manuscript as follows.

The wall loss rates of FA were calculated by introducing FA into a clean chamber and observing the decay. As for $C_9H_{14}O_4$, it was generated during the reaction, and the decay rates were calculated when more than 90% of β-pinene was oxidized and the aerosol concentration kept stable.

5. Line 182– Should "FA and $C_9H_{14}O_4$ (pinic acid and homoterpenylic acid)" actually read "FA and $C_9H_{14}O_4$ (formic acid and homoterpenylic acid)"?

It was revised as:

FA and $C_9H_{14}O_4$ (corresponding to the formula of pinic acid and homoterpenylic acid)

6. Line 187– Figure S5 shows data for $C_9H_{14}O_4$ but not for FA. Why is this? Would the authors be able to provide similar data for FA?

This figure shows the time series of the gas-phase $C_9H_{14}O_4$ in Exp. 298a, 273a, 248a. As the most abundant product during the reaction, the wall loss rates of $C_9H_{14}O_4$ were implemented into the aerosol dynamic model to estimate the vapor wall loss effect.

7. Line 253– The authors should clarify that Figure 4B shows the results from the 'L' experiments at each temperature.

We clarified this as:

Figure 5B shows the temperature dependence of the relative abundance of $C_{16-19}$ dimers at low $[HO_2]/[RO_2]$…

8. Line 271– The uncertainty in the assignment of terpenylic acid, pinic acid and homoterpenylic acid to signals corresponding to C8H12O4 and C9H14O4 should be more explicit here. The authors cannot make a definitive assignment (unless these thermograms were obtained from chemical standards, which I don't believe is true). As such, the chemical names should not just be listed in brackets next to the formulae, but rather should include some phrase indicating a tentative assignment, e.g., "C8H12O4 (corresponding to the formula of terpenylic acid) and C9H14O4 (corresponding to the formulae of pinic acid and homoterpenylic acid)."

We agree on that and have revised them in the manuscript as suggested.

9. Line 283– "…leads to a reduce of more than 40% in total dimers from 248 K to 298 K…" The authors should provide a reference to a figure that illustrates this (e.g., Figure 5)

We added the reference to the figure on this sentence.

The scavenging of SCIs leads to a reduction of more than 40% in total dimers from 248 K to 298 K (Fig. 6)…

10. Line 305– How do the results from Berndt et al. compare to the results from the models presented in this manuscript?

This is a mistake. It should read "not critical for the results." We have corrected this in the manuscript.

11. Line 345– The authors should make it clear that the discussion of ROOR focusses on the gas-phase ROOR, as opposed to the particle-phase dimers that have been discussed up until this point.

We have clarified this in the text as follows.

Where $[RO_2]$ is the concentration of $RO_2$ in the gas phase; [ROOR] is the concentration of dimers formed in the gas phase and quickly partitioned to the particle phase; γ is the branching ratio of dimers formation from $RO_2+RO_2$ reactions. The $RO_2$ concentrations were simulated in the box model for different conditions and are shown in Fig. S19. If $RO_2$ radicals influence dimers formation predominately through $RO_2+RO_2$ reactions, which are second-order reactions and should therefore lead to a linear correlation of dimer signal with $[RO_2]^2$. Zhao et al. (2018) showed a quadratic relationship between the gas-phase signals of dimers and $RO_2$ as evidence of dimers formation from $RO_2+RO_2$ reactions in α-pinene ozonolysis. In this study, we would like to show different impacts of $[HO_2]/[RO_2]$ on dimers formation at different temperatures. Figure 9 shows the correlations between dimers in both phases and simulated $[RO_2]^2$ at different temperatures.

12. Line 362– "It is intriguing to find that at low temperatures, the variation of $[HO_2]/[RO_2]$ has such a big influence on C18 dimers, which are significantly contributed by SCIs reactions." The authors should provide a reference to Figure 7 which illustrates this result.

We added the reference to the figure on this sentence.

It is intriguing to find that at low temperatures, the variation of $[HO_2]/[RO_2]$ has such a big influence on $C_{18}$ dimers (Fig. 8)…

13. Line 367– The authors should state how they obtained the 90% value. Was this obtained from the models or by comparison of particle and gas-phase dimer measurements?

We have clarified this in the revised manuscript as follows.

Figure 10A shows the total abundance of $C_{18}H_{28}O_6$ in the gas and particle phase, and the FIGAERO-CIMS measurements showed that more than 90% of them stayed in the particle phase for all temperatures.

14. Line 369– The authors should outline how they "evaluated the influence of $C_9$-SCIs reactions with CO at lower temperatures.". Was this by looking at nopinone concentrations in the H, M, and L experiments in the same manner as with $C_9H_{14}O_4$ in figure S14?

We have clarified this in the revised manuscript as follows.

We evaluated the influence of $C_9$-SCIs reactions with CO at lower temperatures by comparing the formation of nopinone.

**Minor Comments:**

1. Line 51– "Would this lead to a more important role of SCIs in SOA formation in winter and colder regions of the atmosphere?" It is unusual to phrase this as a question here. If this is an aim of the current study then it should be stated explicitly (e.g. "This study will provide insight into whether this will lead to a more important role of SCIs in SOA formation in winter and colder regions of the atmosphere").

We have rewritten the sentence as suggested.

This study will provide insight into whether this can lead to a different role of SCIs in SOA formation in winter and colder regions of the atmosphere.

2. Line 52– "vital in the atmospheric radical circle, and reactions" should read "atmospheric radical cycle".

We have corrected it according to the reviewer's suggestion.

3. Line 145– The latest MCM version is v3.3.1, not 3.3.2 as stated here. Also, it may be of interest that the MCM is now located at http://mcm.york.ac.uk/, the Leeds website does still work, but the York site is faster.

Thanks for the kind reminder. We have modified this sentence.

The β-pinene reaction mechanism was taken from the Master Chemical Mechanism (MCM) v3.3.1 (http://mcm.york.ac.uk/).

4. Line 190– I don't think that the introduction paragraph here is necessary. At a minimum it should be converted to the future tense for clarity (This discussion will begin with the ... then this will be explored... etc).

We have removed the introduction paragraph.

5. Line 283– "…SCIs leads to a reduce of more than…" should read "…SCIs leads to a reduction of more than…"

We have revised it accordingly.

6. Figure 6– I find the presentation of this data confusing. Particularly the hashed bars, such as $C_{17}H_{22}O_6$, where it is difficult to see the 298K bar. Some potential actions to improve the readability of this figure may be: change the direction of the hashes on one temperature set which would allow the two to be better distinguished, adding a statement to the figure caption indicating that the bars for each temperature are overlaid for each compound, changing the figure colors to provide better contrast, splitting the bars out to not overlap (in a similar fashion to figure 5 or figure 7).

We have replotted this figure to avoid misunderstanding.

[Figure]

**Figure 7. The relative changes of (A) SCIs-controlled and (B) non-SCIs-controlled abundant dimers at 298 K or 248 K versus 273 K (The relative standard deviations are within 25 %, Exp. 298a, 273a, 248a).**

7. Line 327– There should be a new paragraph between these two sentences to indicate that the discussion has shifted from temperature effects to RH effects.

As suggested by the reviewer, we have made it as a new paragraph.

8. Figure 4B– The y-axis label is cut off

We have modified the figure.

In order to consume most β-pinene in the experimental timescale, and to avoid the large wall loss due to the long residence time, the $O_3$ level used in this chamber study was much higher than the typical ambient level. We have clarified this in the experimental section as follows.

It should be noted that the $O_3$ level used in this chamber study was much higher than the typical ambient level for two reasons: first, for better comparison among different conditions more than 90% of β-pinene was expected to be consumed in each experiment; second, a long residence time was avoided to reduce the impact of wall losses.

We also added model simulations at the ambient $O_3$ level, and a discussion on transferring the findings from this chamber study to the real atmosphere was added in section 4.4.

To better transfer the findings in this study to the real atmosphere, box model simulations at ambient $O_3$ level of 40 ppb, and β-pinene of 2 ppb were conducted at 298 K and 248 K. The modeling for more atmospheric conditions focused on $O_3$ reaction, and the OH concentration was set at a low level ($1\times10^4$ molecule cm$^{-3}$). Different HO_2 concentrations of $2\times10^7$ molecule cm$^{-3}$, $1\times10^8$ molecule cm$^{-3}$, and $5\times10^8$ molecule cm$^{-3}$ were used for deriving different $[HO_2]/[RO_2]$ as 0.05–0.1, 0.8–2.5, and >20. Similar mechanisms elaborated in Section 3 were utilized and the proposed reaction coefficients of SCIs were utilized at 248 K. The simulations lasted for 7 days, leading to that the accumulated oxidized β-pinene was 23.5 ppb at 298 K and 12.7 ppb at 248 K. The results showed that with $[HO_2]/[RO_2]$ of 0.05–0.1, accumulated concentrations of dimers formed from C_9-SCIs with RO_2 were $2.26\times10^7$ molecule cm$^{-3}$ at 298 K and $1.18\times10^8$ molecule cm$^{-3}$ at 248 K. With $[HO_2]/[RO_2]$ increasing to 0.8–2.5 and >20, dimers formed from this channel decreased to $6.57\times10^6$ molecule cm$^{-3}$ and $1.28\times10^6$ molecule cm$^{-3}$ at 298 K, and $2.44\times10^7$ molecule cm$^{-3}$ and $4.28\times10^6$ molecule cm$^{-3}$ at 248 K. The atmospheric relevant simulations demonstrated that the variation of temperature and $[HO_2]/[RO_2]$ can have a significant influence on the dimers formation also for ambient $O_3$ levels. It should be noted that in the current simulation, only β-pinene-derived RO_2 were considered, while in the real atmosphere, the C_9-SCIs have opportunities to react with RO_2 formed from other VOCs.

8. Line 196. It is illogical to describe the result first and then introduce Figure 2 in the second sentence.

We have changed the order of these two sentences.

9. Line 207. SOA yield is related to its mass concentration. Both experimental condition and SOA mass concentration in previous studies should be noted.

A summary of the previously reported SOA yield from β-pinene ozonolysis and the corresponding reaction conditions was added to the Supplement.

**Table S1. SOA yields and reaction conditions in β-pinene ozonolysis studies.**

| Temperature (K) | RH (%) | OH scavenger | Aerosol mass (μg·m$^{-3}$) | SOA yield (%) | Reference |
|---|---|---|---|---|---|
| 263–303 | <0.03 | none | 2–284 | 3–39 | von Hessberg et al., 2009 |
| 300 | 22 | none | 156 ± 3 | 18 ± 1 | Xu et al., 2021 |
| 293 | 6.3 | cyclohexane | 174 ± 3 | 17 ± 1 | Lee et al., 2006 |
| 306–307 | – | 2-butanol | 11–19 | 4–8 | Yu et al., 1999 |
| 248–298 | <0.01 | CO | 18–25 | 11–19 | This work |

10. Line 225. The authors showed experiment results about the dependence of SOA formation on $[HO_2]/[RO_2]$ at different temperature. However, discussions about the influencing mechanism of $[HO_2]/[RO_2]$ on SOA yield were not provided in detail. The authors should give more explanations based on MS-measured data and modeled data.

The gas-phase composition from CIMS measurement at different $[HO_2]/[RO_2]$ and temperatures was provided in the Supplement (Fig. S9). We also added a detailed analysis on the effect of $[HO_2]/[RO_2]$ on gas-phase composition and volatility distribution in section 4.1.

$[HO_2]/[RO_2]$ was changed by using different concentrations of CO, resulting in different concentrations of HO_2 radicals. Higher HO_2 concentrations led to a larger sink for RO_2 radicals and consequently a lower RO_2 concentration. Additionally, changing $[HO_2]/[RO_2]$ impacts the branching ratios of product formation from RO_2 reactions. Reaction with HO_2 is a chain termination for RO_2 radicals, leading to the formation of more volatile products, while reactions with RO_2 can be chain

termination or chain propagation processes. Docherty and Ziemann (2003) proposed that the formation of pinic acid in β-pinene oxidation was inhibited by the increasing [HO₂]/[RO₂] as the formation pathway of pinic acid involved a series of RO₂ chain propagation reactions. The inhibition of increasing [HO₂]/[RO₂] on pinic acid formation was also observed in this study for all temperatures. The gas-phase composition shown in Fig. S9 demonstrates that most gas-phase products observed in this study are monomers and smaller molecules, while gas-phase dimers only account for a small fraction. Figure 4 shows the volatility distribution of gas-phase products at different [HO₂]/[RO₂]. Although increasing [HO₂]/[RO₂] promoted the formation of several compounds, the suppression of increasing [HO₂]/[RO₂] on gas-phase products observed from CIMS was more obvious. The relative inhibition of increasing [HO₂]/[RO₂] on gas-phase products was more significant at 273 K and 248 K compared to 298 K. The shift of the volatility classes with decreasing temperature results in the shifts of some gas-phase compounds to lower volatility classes. This could partly explain the larger inhibition effect of increasing [HO₂]/[RO₂] on SOA yield at lower temperatures.

[Figure]

**Figure S9. The averaged CIMS gas-phase mass spectra for all temperatures at different [HO₂]/[RO₂] (Signals are normalized to total gas $C_xH_yO_z$, Exp. 298abc, 273abc, 248abc).**

[Figure]

**Figure 4. Volatility distribution of gas-phase products for different temperatures and [HO₂]/[RO₂] (Exp. 298abc, 273abc, 248abc). Markers are sized by the square root of their relative abundance and colored by the ratio of signals at different [HO₂]/[RO₂] versus signals at low [HO₂]/[RO₂] at each temperature. From left to right, colored bands in the background represent the volatility classes of ULVOC, ELVOC, LVOC, SVOC, and IVOC. These volatility classes are defined for 298 K and shift with temperature according to the Clausius-Clapeyron equation.**

11. Line 229. Could the authors measure the amount of sCI? How did the authors calculate the proportion of the scavenged sCI?

We did not measure the amount of SCIs, and the yield of SCIs from β-pinene ozonolysis reported in previous studies was utilized in the box model. We added explanations on calculating SCIs scavenging proportion in the manuscript.

Since the reaction coefficients of β-pinene-derived $C_9$-SCIs are not clear, it is difficult to calculate the proportion of the scavenged SCIs directly. In SCIs scavenging experiments more than 70% of the gas-phase dimers were diminished for all temperatures, based on which we estimated that more than 70% of $C_9$-SCIs were scavenged.

12. Line 283. Is the proportion of the scavenged sCI determined to be 70% in all experiments?

In SCIs scavenging experiments more than 70% of the gas-phase dimers were diminished for all conditions, based on which we estimated the scavenged $C_9$-SCIs were more than 70%. We have clarified this in the revised manuscript.

13. Line 296. How did the authors measure HCHO and nopinone? Based on the yield of HCHO and nopinone, the authors carried out some updates in the MCM mechanism. Please provide some justifications about MCM updates. Did the author compare the modelled results before and after the update?

HCHO and nopinone were measured by PTR-MS as described in section 2.2. We tried to provide a consolidated description of the model runs and the justification for the updates in section 3.1.

Considering the previously reported Criegee intermediates (CIs) formation in β-pinene ozonolysis and the measurement results from this study, some modifications were applied to the formation of β-pinene-derived CIs, which was elaborated in Section 4.3. By implementing these updates on CIs, the yield of stabilized $CH_2OO$ decreases from 0.15 to 0.1, and the yield of $C_9$-SCIs increases from 0.1 to 0.32. The OH yield from β-pinene ozonolysis decreases slightly from 0.35 to 0.3. It should be noted that for β-pinene SCIs, only reactions with CO, NO, $NO_2$, $SO_2$, and $H_2O$ are concluded in MCM. When CO was used to adjust [HO₂]/[RO₂] by reacting with OH radicals, the possibility of CO reacting with SCIs needed to be estimated. The reaction coefficients of SCIs with CO are usually reported as smaller than $10^{-18}$ $cm^3$ molecule$^{-1}$ s$^{-1}$ (Eskola et al., 2018; Kumar et al., 2014, 2020; Vereecken et al., 2015), and the reaction coefficient of $10^{-18}$ $cm^3$ molecule$^{-1}$ s$^{-1}$ was applied for SCIs reaction with CO in the current model. For better simulating $C_9$-SCIs, the unimolecular reaction with a coefficient of 75 s$^{-1}$ and reactions with HO₂ and RO₂ with coefficients of $2\times10^{-11}$ $cm^3$ molecule$^{-1}$ s$^{-1}$ were implemented in the model, which was elaborated in Section 4.4. These modifications showed a limited impact on HO₂ and RO₂ concentrations. The modeling results shown below were derived by implementing these updates.

14. Lines 346. RO₂ could participate in dimer formation via its self or cross reaction. How did the author consider the RO₂ self-reaction?

In this manuscript, RO₂+RO₂ reactions include the self- and cross-reactions of RO₂ radicals. We have clarified this where "RO₂+RO₂ reactions" was used for the first time.

As for RO₂+RO₂ reactions, which include the self- and cross-reactions of RO₂ radicals, the rate coefficients vary over a wide range…

15. Line 692. There is no experimental information in the caption.

We have added the experimental information in the caption.

16. Many sentences in this manuscript are too long to obtain useful information. Please double check.

We greatly thank the reviewer for the kind reminder. We have checked and modified these long sentences in the revised manuscript.

---

## Author Comment (AC3)

**Response to reviewers' comments on "Impact of temperature on the role of Criegee intermediates and peroxy radicals in dimers formation from β-pinene ozonolysis"**

The authors greatly thank the reviewers for the careful review of our manuscript and the valuable feedback. All the comments are addressed point by point, with our responses in blue, and the corresponding revisions to the manuscript in red. All updates are marked in the revised manuscript. According to the comments, we have added new analysis to strengthen our work.

**Response to Reviewer #1**

Gong et al. studied the SOA formation of  $\beta$ -pinene ozonolysis at 298 K, 273 K and 248 K. Coupling the laboratory measurement and chemical modelling, the authors provide insights into the mechanisms of dimers formation in the  $\beta$ -pinene ozonolysis SOA at three studied temperatures. Considering the importance of SOA formation in winter or cold regions or high altitude, the topic is of high interest to the research community in atmospheric science. The study was well designed, but only a subset of data was analyzed and presented. The current discussion is mostly limited to the dimers in the particle phase. I would highly encourage the authors to carry out more comprehensive data analysis for the gas- and particle-phase data before they reach the conclusions. I would support the final publication after addressing my comments below.

**Major Comments**

 SOA studies at low temperatures are increasingly popular (Huang et al., 2018; Ye et al., 2019; Simon et al., 2020; Gao et al., 2022; Gao et al., 2023), of which the findings inform the SOA formation at high altitudes or colder regions. In particular, several KIT studies (Huang et al., 2018; Gao et al., 2022; Gao et al., 2023) which used the same chamber as this study have reported the effect of temperature on the formation of biogenic SOA and their physicochemical properties. The authors should briefly summarize the findings of the previous studies at least from KIT and then discuss what new values this study is bringing.

We have summarized the previous findings on the temperature effect on SOA, and this part is included in the introduction section as follows.

Several studies reported higher SOA yields at lower temperatures in  $\alpha$ -pinene oxidation (Jonsson et al., 2008; Pathak et al., 2007, 2008; Saathoff et al., 2009). In recent years, more attention has been paid to the temperature impact on aerosol constituents and physicochemical properties. Ye et al. (2019) and Simon et al. (2020) reported highly oxygenated molecules (HOMs) were less abundant at lower temperatures in  $\alpha$ -pinene oxidation due to the positive temperature dependence of autoxidation reaction (Praske et al., 2017), however, the reduction of the saturation vapor pressure at lower temperatures counteracted the chemical effect on new particle formation. Kristensen et al. (2017) reported suppressed formation of dimers at subzero temperatures. Huang et al. (2018) studied the interactions between particle composition and viscosity at 223 K. Gao et al. (2021, 2023) investigated the temperature effect on the composition and volatility of aerosols from  $\beta$ -caryophyllene oxidation. These studies indicate that besides the impact on volatility and partitioning, temperature impacts the chemical reaction mechanism and product formation. Although the temperature impact on HOMs formation was studied, the understanding of the temperature impact on the formation pathways of other important SOA constituents such as dimers is limited. This study tries to bring some new insights into the reaction mechanisms of two kinds of reactive intermediates in the atmosphere: stabilized Criegee intermediates (SCIs) and organic peroxy radicals (RO2) at lower temperatures, and their further impacts on dimers and SOA formation.

2. The  $O_3$  concentration used in the experiments was approximately 1 ppm, which was higher than the typical ambient level. How are the findings of the study applicable to the real atmosphere? Please give a justification. To further strengthen the implications of the work, model simulations should be also carried out for a scenario of ambient level of  $O_3$ .

In order to consume most  $\beta$ -pinene in the experimental timescale, and to avoid the large wall loss due to the long residence time, the O3 level used in this chamber study was much higher than the typical ambient level. We have clarified this in the experimental section as follows.

It should be noted that the  $O_3$  level used in this chamber study was much higher than the typical ambient level for two reasons: first, for better comparison among different conditions more than 90% of  $\beta$ -pinene was expected to be consumed in each experiment; second, a long residence time was avoided to reduce the impact of wall losses.

We also added model simulations at the ambient  $O_3$  level, and a discussion on transferring the findings from this chamber study to the real atmosphere was added in section 4.4.

To better transfer the findings in this study to the real atmosphere, box model simulations at ambient O3 level of 40 ppb, and  $\beta$ -pinene of 2 ppb were conducted at 298 K and 248 K. The modeling for more atmospheric conditions focused on O3 reaction, and the OH concentration was set at a low level (1×104 molecule cm-3). Different HO2 concentrations of 2×107 molecule cm-3, 1×108 molecule cm-3, and 5×108 molecule cm-3 were used for deriving different [HO2]/[RO2] as 0.05–0.1, 0.8–2.5, and >20. Similar mechanisms elaborated in Section 3 were utilized and the proposed reaction coefficients of SCIs were utilized at 248 K. The simulations lasted for 7 days, leading to that the accumulated oxidized  $\beta$ -pinene was 23.5 ppb at 298 K and 12.7 ppb at 248 K. The results showed that with [HO2]/[RO2] of 0.05–0.1, accumulated concentrations of dimers

formed from C9-SCIs with RO2 were  $2.26 \times 10^7$  molecule cm-3 at 298 K and  $1.18 \times 10^8$  molecule cm-3 at 248 K. With [HO2]/[RO2] increasing to 0.8–2.5 and >20, dimers formed from this channel decreased to  $6.57 \times 10^6$  molecule cm-3 and  $1.28 \times 10^6$  molecule cm-3 at 298 K, and  $2.44 \times 10^7$  molecule cm-3 and  $4.28 \times 10^6$  molecule cm-3 at 248 K. The atmospheric relevant simulations demonstrated that the variation of temperature and [HO2]/[RO2] can have a significant influence on the dimers formation also for ambient O3 levels. It should be noted that in the current simulation, only  $\beta$ -pinene-derived RO2 were considered, while in the real atmosphere, the C9-SCIs have opportunities to react with RO2 formed from other VOCs.

3. If the instrumentation was operated at a temperature which was higher than that in the AIDA chamber, potential losses of particle-phase compounds due to evaporation could occur in the tubing. Please justify how the temperature difference affected the observed gas and particle components in detail.

This is an issue that needs to be considered for low temperature experiments. We added a discussion on this issue in the experimental section.

Since all the instruments and the filter sampling were operated at  $295\pm2$  K, the influence of the temperature difference to the simulation chamber needs to be considered (Gao et al., 2023). For the online measurement instruments, a bypass flow was added to reduce the residence time in the sampling tubes to be less than 10 s. Such a short residence time avoids significant particle evaporation and diminishes artifacts on online measurements. For particles collected on PTFE filters, which were later analyzed by FIGAERO-CIMS, the sampling time of each filter was 15–20 minutes. Due to the short residence time of about 1 s in the sampling line the filter was significantly cooled for low temperature experiments. Before storing the filter samples in a freezer at 253 K there were also 5–10 minutes of handling time. Hence, we cannot rule out that some particulate compounds could evaporate during the sampling, resulting in a potential underestimation of some more volatile compounds in the particle phase for low temperature experiments. However, considering the dry conditions of our experiments, substantial evaporation of (semi-)volatile compounds from particle phase should be hindered due to the high viscosity of the particles.

4. Lines 220–231 and Lines 242–244: The authors are encouraged to provide more insights into how [HO2]/[RO2] affect the SOA formation at different temperatures. Given the gas-phase CIMS measurement was available in the study, the authors should provide detailed analysis on how different [HO2]/[RO2] affects the gas phase composition, volatility distributions and consequently SOA formation.

The gas-phase composition from CIMS measurement for different  $[HO_2]/[RO_2]$  and temperatures was provided in the Supplement (Fig. S9). We also added a detailed analysis on the effect of  $[HO_2]/[RO_2]$  on gas-phase composition and volatility distribution in section 4.1.

 $[HO_2]/[RO_2]$  was changed by using different concentrations of CO, resulting in different concentrations of HO2 radicals. Higher HO2 concentrations led to a larger sink for RO2 radicals and consequently a lower RO2 concentration. Additionally, changing  $[HO_2]/[RO_2]$  impacts the branching ratios of product formation from RO2 reactions. Reaction with HO2 is a chain termination for RO2 radicals, leading to the formation of more volatile products, while reactions with RO2 can be chain termination or chain propagation processes. Docherty and Ziemann (2003) proposed that the formation of pinic acid in  $\beta$ -pinene oxidation was inhibited by the increasing  $[HO_2]/[RO_2]$  as the formation pathway of pinic acid involved a series of RO2 chain propagation reactions. The inhibition of increasing  $[HO_2]/[RO_2]$  on pinic acid formation was also observed in this study for all temperatures. The gas-phase composition shown in Fig. S9 demonstrates that most gas-phase products observed in this study are monomers and smaller molecules, while gas-phase dimers only account for a small fraction. Figure 4 shows the volatility distribution of gas-phase products at different  $[HO_2]/[RO_2]$  on gas-phase products observed from CIMS was more obvious. The relative inhibition of increasing  $[HO_2]/[RO_2]$  on gas-phase products was more significant at 273 K and 248 K compared to 298 K. The shift of the volatility classes with decreasing temperature results in the shifts of some gas-phase compounds to lower volatility classes. This could partly explain the larger inhibition effect of increasing  $[HO_2]/[RO_2]$  on SOA yield at lower temperatures.

Figure S9. The averaged CIMS gas-phase mass spectra for all temperatures at different [HO2]/[RO2] (Signals are normalized to total gas CxHyOz, Exp. 298abc, 273abc, 248abc).